# Specific 3-*O*-sulfated heparan sulfate domains regulate salivary gland basement membrane metabolism and epithelial differentiation

Vaishali N. Patel [1]✉, Marit H. Aure [1], Sophie H. Choi[1], James R. Ball[1], Ethan D. Lane [1], Zhangjie Wang[2,3], Yongmei Xu[2], Changyu Zheng[4], Xibao Liu[5], Daniel Martin [6], Jillian Y. Pailin[1], Michaela Prochazkova[7], Ashok B. Kulkarni[7], Toin H. van Kuppevelt[8], Indu S. Ambudkar[5], Jian Liu[2] & Matthew P. Hoffman [1]✉

Heparan sulfate (HS) regulation of FGFR function, which is essential for salivary gland (SG) development, is determined by the immense structural diversity of sulfated HS domains. 3-*O*-sulfotransferases generate highly 3-*O*-sulfated HS domains (3-*O*-HS), and *Hs3st3a1* and *Hs3st3b1* are enriched in myoepithelial cells (MECs) that produce basement membrane (BM) and are a growth factor signaling hub. *Hs3st3a1;Hs3st3b1* double-knockout (DKO) mice generated to investigate 3-*O*-HS regulation of MEC function and growth factor signaling show loss of specific highly 3-*O*-HS and increased FGF/FGFR complex binding to HS. During development, this increases FGFR-, BM- and MEC-related gene expression, while in adult, it reduces MECs, increases BM and disrupts acinar polarity, resulting in salivary hypofunction. Defined 3-*O*-HS added to FGFR pulldown assays and primary organ cultures modulates FGFR signaling to regulate MEC BM synthesis, which is critical for secretory unit homeostasis and acinar function. Understanding how sulfated HS regulates development will inform the use of HS mimetics in organ regeneration.

FGFR signaling is required for the development, function, and regeneration of a wide range of organs, including salivary glands (SG)[1,2]. The essential role of FGFR signaling in SGs is highlighted by mutations in FGF10 or FGFR2b, which lead to two rare genetic disorders in humans; aplasia of lacrimal and salivary glands (ALSG: MIM #180920) and lacrimo-auriculo-dento-digital syndrome (LADD: MIM #149730)[3]. In addition, SG dysfunction due to irradiation damage following head and neck cancer therapy or after autoimmune destruction in Sjögren's disease remains a major clinical challenge, and FGFR signaling is an attractive target for regenerative strategies[4,5]. The functions of many FGFs are regulated by heparan sulfate (HS) either by increasing the affinity of FGF/FGFR signaling complexes or by acting as a reservoir for growth factors on the cell surface or in the basement membrane (BM), a

[1]Matrix and Morphogenesis Section, National Institute of Dental and Craniofacial Research, NIH, DHHS, Bethesda, MD, USA. [2]Division of Chemical Biology and Medicinal Chemistry, Eshelman School of Pharmacy, University of North Carolina, Chapel Hill, NC, USA. [3]Glycan Therapeutics Corp, Raleigh, NC, USA. [4]Translational Research Core, National Institute of Dental and Craniofacial Research, NIH, DHHS, Bethesda, MD, USA. [5]Secretory Physiology Section, National Institute of Dental and Craniofacial Research, NIH, DHHS, Bethesda, MD, USA. [6]NIDCD/NIDCR Genomics and Computational Biology Core, National Institute of Dental and Craniofacial Research, NIH, DHHS, Bethesda, MD, USA. [7]Functional Genomics Section, National Institute of Dental and Craniofacial Research, NIH, DHHS, Bethesda, MD, USA. [8]Department of Biochemistry, Radboud University Medical Center, Nijmegen, The Netherlands. ✉e-mail: vapatel@nih.gov; mhoffman@nih.gov

specialized extracellular matrix (ECM) that separates epithelia from stroma.

BMs are highly dynamic during development and important for cell polarity, migration and signaling[6,7]. BM metabolism, a balance of synthesis and proteolysis, is essential for organogenesis and generates BM cleavage products with signaling functions as well as releasing growth factors stored in the BM[8]. BMs are composed of laminin and collagen IV networks that are independent but linked by HS proteoglycans (HSPGs) such as perlecan and collagen 18[9]. We reported that matrix metalloproteinase (MMP)-dependent collagen IV NC1 domain release increases the synthesis of *Col4a2*, *Fgfr1b*, *Fgfr2b* and *Mmp15* expression via β1 integrin and PI3K-AKT signaling, thus coordinating BM metabolism with epithelial proliferation during SMG development[10]. BM assembly is also dependent on HS sulfation as HSPGs interact with the independent networks of laminins and collagen IV to play a major role in the assembly and structure of BMs[11,12].

Although HS is ubiquitously expressed in cells and in ECM, including the BM, the complex regulation of growth factor binding and signaling is determined by the immense structural diversity of highly sulfated HS domains attached to HSPGs. 3-*O*-sulfation creates the most highly sulfated but rare 3-*O*-sulfated HS domains (3-*O*-HS), which are generated by the largest family of HS biosynthetic enzymes in a spatially and temporally regulated way in different cell types and tissue environments[13]. These enzyme isoforms can generate two types of 3-*O*-HS with specific activities; Hs3st1-like, which binds antithrombin[14], and Hs3st3-like, which binds HSVgD1[15,16]. Seven enzyme isoforms generate the 3-*O*-sulfated domain, and functional redundancy exists among them. The submandibular gland (SMG) epithelium only expresses four of the seven *Hs3st* sulfotransferases (*Hs3st1*, *Hs3st3a1*, *Hs3st3b1*, *Hs3st6*) in a spatial and temporal manner[15,17]. The expression of fewer enzyme isoforms reduces potential functional redundancy in the tissue making the SMG a useful model to investigate 3-*O*-sulfation. Major issues recently overcome were the biochemical analysis of 3-*O*-HS since 3-*O*-sulfation can cause resistance to enzyme digestion, and how to directly measure whether a genetic deletion of specific 3-*O*-sulfotransferase reduces 3-*O*-HS in tissue[18,19]. Increasing the concentration of heparinase enzymes has also recently enabled the digestion of the more resistant structures to 3-*O*-sulfated disaccharides[20]. In addition, recent chemoenzymatic synthesis of defined 3-*O*-HS structures allows us to investigate their function. These advances allow more specific probing into the study of the 3-*O*-sulfated structure and function relationship of HS.

In SG, *Hs3st3a1* and *Hs3st3b1* are enriched in MECs[17], which wrap around secretory acinar cells and can act as progenitors during development, homeostasis, and regeneration[21–24]. MEC produce BM proteins such as laminins, fibronectins, HSPGs and collagen IV, and are known to be regulators of tissue polarity[25,26]. Preclinical models show that MECs have bipotent progenitor potential following severe damage, while during homeostasis, they are unipotent and self-maintained[21,27]. We recently showed that nerve growth factor (NGF) drives MEC differentiation during development, and upregulated neurotrophin signaling in human MEC after irradiation for cancer is associated with stress-induced plasticity and lack of regeneration[28]. MECs arise from SG endbud progenitors, which transition from a keratin14 (Krt14)+ multipotent state via two bipotent states, and to a unipotent state during homeostasis[29]. In postnatal glands, MECs act as a signaling hub, expressing FGF7, which activates an acinar transcriptional program[28] involving both FGFR2[1] and Kras[29] signaling to drive acinar differentiation. Single *Hs3st3a1* or *Hs3st3b1* knockout mice have subtle secretory phenotypes, suggesting that the loss of only one *Hs3st3* enzyme is functionally compensated for by another Hs3st3 enzyme.

Here, we generated 3-*O*-sulfotransferase DKO mice to study the function of 3-*O*-sulfation in vivo. We confirmed the loss of specific 3-*O*-sulfated domains using mass spectrometry and analyzed gene expression at multiple stages of SMG development, myoepithelial cell cultures and using defined chemoenzymatically synthesized HS in explant culture to interrogate 3-*O*-sulfated-HS function. We observe a reduction in MEC and ducts in the adult gland and a proportional increase in acinar cells. The secretory units, which contain acinar cells surrounded by MECs and BM, are disrupted, leading to gland hypofunction. We discover that loss of these highly sulfated domains disrupts FGFR signaling, BM metabolism, MEC, and duct development, resulting in secretory unit dysfunction.

## Results

### Generation of the *Hs3st3a1;Hs3st3b1* DKO mice

We previously generated *Hs3st3a1* and *Hs3st3b1* knockout (KO) mice[17] and have also obtained *Hs3st1* KO mice, all on a C57BL/6 background[30,31]. Both *Hs3st3a1* and *Hs3st3b1* genes are on Chromosome 11 about 511 kb apart, likely a result of a gene duplication event, therefore crossing the single knockouts of *Hs3st3a1* and *Hs3st3b1*[17] was not feasible. To generate DKO mice, zinc finger nuclease (ZFN) targeting of mRNAs to exon 1 of *Hs3st3a1* were microinjected into single-cell *Hs3st3b1* heterozygous embryos and transferred to pseudopregnant recipients. Of 119 live births, cleaved DNA was detected from 22 pups by T7 endonuclease assays. Sequencing PCR products from tail snip DNA identified seven founders with frameshift deletions that introduced a premature stop codon. These founders were bred with wildtype (WT) mice, and three founders were selected with deletion of both *Hs3st3a1* and *Hs3st3b1* segregated on the same allele. Founders B9, B14 and D6 had 10, 26 and 22 base pair deletions, respectively, in *Hs3st3a1* (Supplementary Fig. 1a). These mice were sequenced to confirm the nucleotides were deleted in *Hs3st3a1* (Supplementary Fig. 1b) and PCR detection of the LacZ cassette, to determine loss of *Hs3st3b1* (Supplementary Fig. 1c). The *Hs3st3* heterozygous mice were intercrossed, and phenotypic characterization was performed on offspring. The DKO mice were viable, fertile and obtained at the expected mendelian ratios with Chi-square analysis showing no significant difference (Fig. 1a). In contrast to the single KO strains[17], DKO mice were ~25% smaller in size than their WT littermates (Fig. 1b). Further, gross anatomic analyses of adult SMGs showed relatively larger glands in DKO compared to WT when normalized to the body weights (Fig. 1b). Taken together, this suggested that DKO mice have less functional redundancy than single KO strains.

### DKO SMGs have loss of highly sulfated 3-*O*-sulfated tetrasaccharides

The genetic deletion of two sulfotransferases within a large family that may have functional redundancy highlighted the need to determine whether 3-*O*-sulfation was reduced in DKO HS. Disaccharide analysis after bacterial heparin lyase depolymerization leaves enzyme-resistant 3-*O*-sulfated tetrasaccharides that until recently could not be resolved due to lack of tetrasaccharide standards, but recent advances in LC-MS/MS methods have overcome this barrier[18–20]. We analyzed both non-3-*O*-sulfated disaccharides and 3-*O*-sulfated tetrasaccharides in the male SMGs from DKO and the single KOs of *Hs3st1*, *Hs3st3a1* and *Hs3st3b1*. (Fig. 1c, d and Supplementary Fig. 1d, e). Disaccharide analysis showed a significant increase in ΔUA2S-GlcNS and an increasing trend for ΔUA-GlcNS, while other disaccharides showed no differences in the DKO SMGs compared to the WT (Fig. 1c). Disaccharide analysis in the single knockouts did not show any differences compared to the WT control (Supplementary Fig. 1d).

Interestingly, tetrasaccharide analysis resolved 5 different 3-*O*-sulfated tetrasaccharides in WT HS. One tetrasaccharide, ΔUA-GlcNS6S-IdoA2S-GlcNS3S6S, was not detected in the DKO, *Hs3st3a1* KO and *Hs3st3b1* KO (Fig. 1d, Supplementary Fig. 1e). This most highly sulfated tetrasaccharide is predicted to be the product of HS3ST3 enzymes. In addition, ΔUA-GlcNS-IdoA2S-GlcNS3S, which is also predicted to be a product of HS3ST3 enzymes, was not significantly

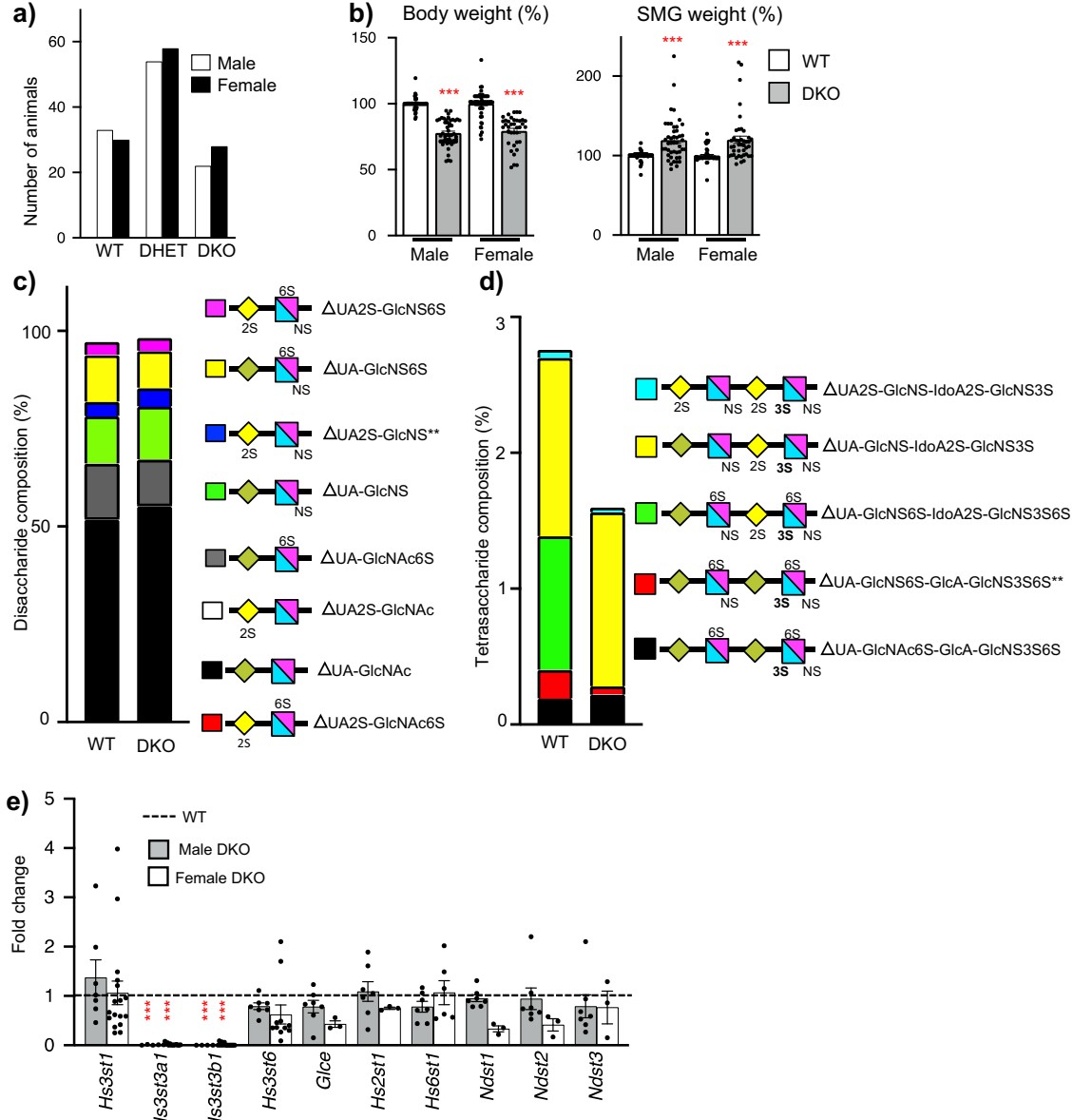

**Fig. 1 | DKO mice are viable but are smaller, with larger SMGs, and have a reduction in specific 3-O-sulfated tetrasaccharides compared to WT. a** Data shown are the number of mice of a given genotype for each sex derived from 25 independent litters obtained from heterozygous crosses. Chi-square analysis shows no significant difference from the expected ratios. **b** Body weight and SMG weight of adult male and female mice normalized to body weight and WT. Error bars are SM. $n = 65$ (WT-male), 51 (DKO-male), 49 (WT-female) and 35 (DKO-female) mice. SMG weight: $n = 63$ (WT-male), 46 (DKO-male), 47 (WT-female) and 42 (DKO-female). Unpaired two-tailed $t$-test. ***$p < 0.0001$, compared to WT. **c, d** Mass spec analysis of HS disaccharides and tetrasaccharides in adult male SMGs. Analysis of HS disaccharide (**c**) and tetrasaccharide (**d**) composition between WT and DKO SMGs. All data shown is averaged and presented as stacked bar graphs. $n = 3$ biological repeats for each group. Paired two-tailed $t$-test compared to

WT, ΔUA2S-GlcNS **$p = 0.004$, and ΔUA-GlcNS6S-GlcA-GlcNS3S6S **$p = 0.0057$. ΔUA2S-GlcNAc6S not detected in WT and DKO SMGS. **e** Gene expression changes in HS biosynthetic enzymes in both male and female DKO SMGs normalized to the respective male or female WT SMGs and $Rps29$. Error bars are SM. $n = 7$ for WT-M and DKO-M SMGs. $n = 3$ for WT-F ($Glce$ and $Hs2st1$) and DKO-F ($Glce, Hs2st1, Ndst1, Ndst2$ and $Ndst3$), $n = 6$ for WT-F ($Ndst2, Ndst3$) and DKO-F ($Hs6st1$), $n = 10$ for WT-F ($Hs3st3b1$ and $Hs3st6$), $n = 11$ DKO-F ($Hs3st6$), $n = 14$ DKO-F ($Hs3st3a1$ and $Hs3st3b1$), $n = 15$ WT-F ($Ndst1$), $n = 16$ WT-F ($Hs3st1$ and $Hs3st3a1$), $n = 17$ DKO-F ($Hs3st1$) and $n = 18$ DKO-F ($Hs6st1$). Unpaired two-tailed $t$-test compared to WT, ***$p < 0.0001$ for $Hs3st3a1$ and $Hs3st3b1$ in DKO-male and DKO-female. Source data are provided as a Source Data file.

decreased in the DKO, $Hs3st3a1$, or $Hs3st3b1$ KO SMGs. The tetrasaccharide, ΔUA-GlcNS6S-GlcA-GlcNS3S6S, containing 3-O-sulfation correlating to the antithrombin binding site proposed to be generated by HS3ST1 significantly decreased in the DKO HS (Fig. 1d), but not in the $Hs3st3a1$, $Hs3st3b1$ and $Hs3st1$ KO SMGs compared to WT. Interestingly, ΔUA-GlcNAc6S-GlcA-GlcNS3S6S was not detected in the SMGs of $Hs3st1$ KO SMGs (Supplementary Fig. 1e), whereas no differences were detected in Hs3st3 KO mice.

Direct comparison of male and female SMGs within the same analytical batches showed overall more differences in the HS composition of the disaccharides than the tetrasaccharides between WT males and WT females. Specifically, UA-GlcNAc, UA2S-GlcNAc, and UA2S-GlcNAc6S disaccharides were increased, whereas UA-GlcNAc6S and UA-GlcNS6S were reduced in WT female SMGs compared to WT male SMGs (Supplementary Fig. 1f). Only tetrasaccharide UA2S-GlcNS-IdoA2S-GlcNS3S was different in the SMGs of the two sexes

(Supplementary Fig. 1g). The DKO SMGs were more similar in their HS with only UA2S-GlcNAc6S and UA2S-GlcNS being different. Further work is needed to fully understand the biological significance of these differences, however, this analysis showed that both males and females have altered HS composition in the SMGs of DKO mice.

Quantitative PCR analysis in the adult DKO SMGs showed, as expected, no expression of *Hs3st3a1* and *Hs3st3b1*, while other HS enzymes in DKO SMGs were comparable to the WT (Fig. 1e). Both WT and DKO adult SMG had no detection of *Hs3st2*, *Hs3st4*, *Hs3st5*, *Hs6st3*, and *Ndst4*.

Taken together, these analyses confirm there is a significant decrease in both ΔUA-GlcNS6S-IdoA2S-GlcNS3S6S and ΔUA-GlcNS6S-GlcA-GlcNS3S6S tetrasaccharides generated by Hs3st3 enzymes without compensatory upregulation of other biosynthetic enzymes in DKO SMG.

## Adult DKO SMGs have a disrupted epithelial compartment

The murine SMG is sexually dimorphic; males have larger granular ducts with prominent pink granular cytoplasm compared to those of the female (Fig. 2a, arrows). When compared to WT, the gross morphology of DKO SMG in both sexes showed a reduction in granular ducts and an increase in acinar cells per area (Fig. 2a), suggesting the epithelial progenitor niche may be affected.

To identify transcriptional changes in the DKO adult SMGs, RNAseq analysis was performed with selection parameters including a two-fold change in expression with an adjusted *p*-value of 0.05. We identified 451 and 621 differentially expressed genes (DEGs) between DKO and WT littermates in female and male SMGs, respectively (Supplementary Data 1). Of these, 111 DEGs were common to male and female DKO compared to the WT. Mining previous scRNAseq analysis of adult SMGs[24], we generated cell-type-specific heatmaps of DEGs in male and female SMGs (Fig. 2b, Supplementary Fig. 2a). Consistent with gross morphology, the majority of upregulated genes were acinar markers. *Aqp5*(Aquaporin 5), *Agt* (angiotensinogen), *Mucl1* (Mucin-like 1) and *Agr2* (protein disulfide isomerase), were all increased in female DKO, whereas *Lpo*, *Mucl2* and *Caecam1* were increased in male (Fig. 2b, c). There was decreased ductal gene expression of the entire *Mus musculus* species-specific kallikrein-related peptidases (KLK) subfamily Klk1b (Fig. 2b, c), which are serine proteases expressed by granular ducts[32,33]. Other downregulated DEGs granular duct genes were growth factors *Egf* and *Ngf*. In general, duct specific genes were reduced more in male DKO SMGs, consistent with the predominance of granular ducts in the male murine SMG. Surprisingly, we detected a reduction in MEC genes despite secretory units being proportionally increased in DKOs. *Hs3st3a1* and *Hs3st3b1* are enriched in MECs[17], suggesting a cell-autonomous role for 3-*O*-sulfated HS in MEC differentiation and/or maintenance.

Validation of RNAseq by qPCR (Fig. 2c) confirmed the overall trends measured in male and female SMGs. Examples of MEC genes that were reduced in expression include those for contractile proteins such as *Acta2* (Smooth muscle actin), *Cnn1* (calponin 1), *Tagln* (transgelin), *Myh11* (myosin heavy chain 11), *Myl9* (myosin light chain 9), *Mylk* (myosin light chain kinase) and *Tpm2* (tropomyosin 2), as well as the BM component, *Lama2* (laminin alpha 2). In addition, *Krt14* and *Postn* (a secreted ECM protein involved in integrin-mediated cell migration), which are expressed in both MECs and basal ducts, were also reduced. DEGs included growth factor genes such as *Ctgf*, *Fgf1*, *Fgf9*, *Pdgfa*, *Tgfb2*, and *Kitl*, which were all confirmed by qPCR in the male and/or female DKO SMG.

We performed KEGG pathway analysis of the common 111 DEGs, and in general, this reflected reduced duct and MEC gene signatures. The pathways affected highlight the role of MECs as signaling hubs[28], producing growth factors that signal via PI3K-AKT (30 genes), MAPK (23 genes) and adrenergic signaling pathways (12 genes) (Supplementary Table 1a, b). In addition, genes associated with focal adhesions

(21 genes), ECM-receptor interactions (10 genes), cell adhesion molecules (19 genes) also reflect the MEC function of producing basement membrane components and HSPGs, which influences cell-matrix interactions.

We further confirmed the reduction of ducts and MECs and relative increase in acinar cells by immunostaining of acinar cell membranes (Acinar1, Ac1), MEC and basal ducts (keratin 14, K14), granular ducts (mucin13, M13 and EGF), and MECs (smooth muscle actin, SMA) (Fig. 2d). Quantitation confirmed an increase in acinar cells (Ac1), whereas MEC and basal ducts (K14) were reduced although it was apparent the MEC staining was reduced compared to basal duct staining. There was also a reduction in granular duct staining (M13 and EGF) (Fig. 2e). To confirm the reduction in MECs, we also stained with SMA, a more specific MEC marker, which also showed reduced MEC staining.

Together, these data show that the loss of *Hs3st3a1* and *Hs3st3b1* affects the epithelial progenitor niche, which leads to a disrupted epithelial compartment, particularly the secretory units. Despite the increase in acinar cells, there was a reduction in MECs, which are contractile and produce growth factors and BM. The loss of *Hs3st3a1* or *Hs3st3b1* alone previously did not affect the MECs, acinar or basal ducts cells[17]. Therefore, we next investigated how changes in HS influence FGFR signaling and secretory unit function in DKO SMGs.

## The DKO BM has increased binding of FGF10/FGR2b complex and increased staining of other BM components

To determine whether the loss of 3-*O*-HS domains alters the BM or cell surface HS, we performed the ligand and carbohydrate engagement (LACE) assay. This assay is a sensitive tool using recombinant FGF/FGFR complexes to probe HS structures on cells and BM of tissue, as HS is required to form high-affinity HS/FGF/FGFR signaling complexes. Surprisingly, the DKO had increased binding of the FGF10/FGFR2b complex surrounding acinar structures and ducts, overlapping with increased type-IV collagen (ColIV) staining in the BM compared to WT (Fig. 3a and Supplementary Fig. 3a, b). The LACE assay is HS-dependent as treatment with heparinase III, abolished FGF10/FGFR2b binding, whereas chondroitinase ABC enzyme treatment had no effect (Supplementary Fig. 3c). We also compared binding of other FGF/FGFR combinations by LACE and found that both FGF1/FGFR2b and FGF7/FGFR2b binding, increased in the DKO, whereas FGF1/FGFR1b was similar to WT (Supplementary Fig. 3d, e). Overall, these data confirm that the HS in the DKO increases FGF/FGFR2b binding, which may reflect increased FGFR signaling in vivo. The increased LACE and ColIV staining suggest potential feedback to increase BM synthesis, reflecting an alteration in BM metabolism. In contrast to the DKO, no changes in the binding of FGF10/FGFR2b complex to the endogenous HS or COLIV staining were previously detected in *Hs3st3a1* or *Hs3st3b1* single KO SMGs[17].

To further investigate whether HSPGs were affected, we stained with the 3G10 antibody, which binds an HS stub, a neo-epitope generated by heparinase III pre-treatment. 3G10 staining reflects the amount of HSPG core proteins and/or the number of HS attachment sites per molecule[34], and 3G10 staining also increased surrounding the mucin10+ acinar structures in the DKO (Fig. 3b, c). E-cadherin (ECAD) antibodies stained all epithelium and were similar in WT and DKO. These data may suggest that HSPGs in the BM are increased in the DKO SMG (Fig. 3c), which is supported by 3G10 Western blot analysis of tissue lysates treated with heparinase III and chondroitinase ABC enzymes, revealing increased high molecular weight core HSPGs (Fig. 3d). Although, RNAseq did not identify changes in BM (*Hspg2*, *Agrn*, and *Col18a1*) or cell surface (*Sdc1* and *Sdc4*) HSPG transcripts in the DKO. Taken together, the increased LACE and 3G10 antibody staining suggested there may potentially be increased FGF/FGFR signaling in the DKO.

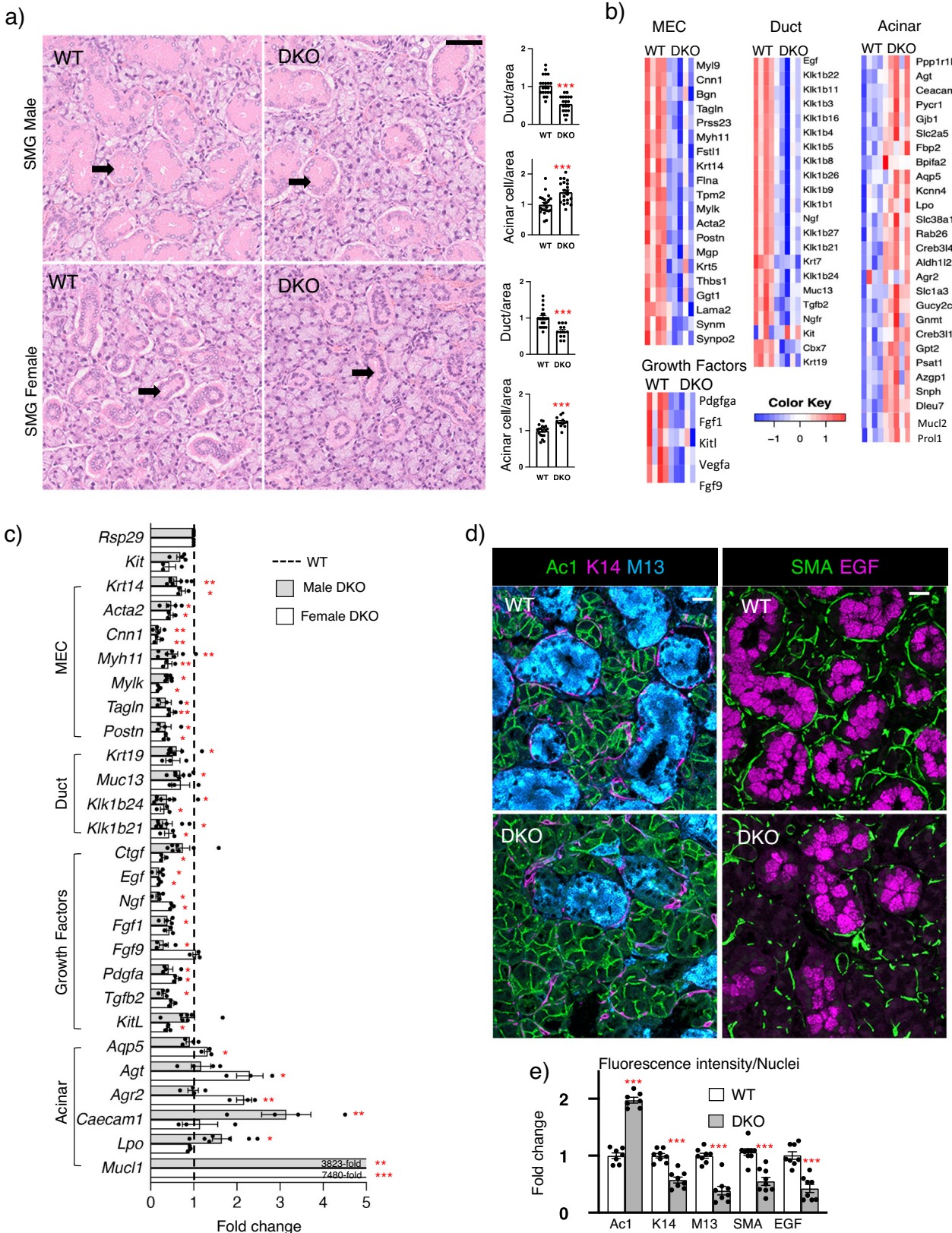

## MAPK signaling is increased in the acinar cells in male SMGs

We next investigated whether MAPK phosphorylation, which is downstream of FGFR signaling, was affected. Western blotting of SMG lysates detected increased pERK normalized to total ERK in male SMGs (Fig. 4a). Although, pERK was not significantly increased in the female DKO SMGs lysates (Supplementary Fig. 4a). The localization of pERK in male DKO SMG was analyzed using immunostaining and showed an increase in pERK staining in the DKO acinar cells (Fig. 4b and Supplementary Fig. 4b). The staining specificity was confirmed by treating sections with phosphatase, which abolishes staining (Fig. 4b).

**Fig. 2 | DKO SMGs have fewer ducts and MECs and relatively more acini compared to WT, reflected in RNAseq analysis, qPCR and immunostaining.**
**a** Representative H&E images of adult male (M) and female (F) WT and DKO SMGs. Arrows point to granular ducts. Scale bar = 50 μm. Ducts and acinar cells quantified per area normalized to the WT. $n = 4$ (WT-M, DKO-M, and WT-F) and $n = 3$ (DKO-F) SMGs with multiple areas quantified (WT-M: $n = 21$, DKO-M: $n = 20$, WT-F: $n = 19$, DKO-F: $n = 12$). Unpaired two-tailed $t$-test. DKO-M acini \*\*\*$p = 0.0004$, DKO-M duct \*\*\*$p < 0.0001$, DKO-F acini \*\*\*$p = 0.0004$, and DKO-F duct \*\*\*$p < 0.0001$ compared to WT. **b** RNAseq analysis of DKO SMGs identifies reduced MEC and duct genes with the increase of some acinar genes. Heatmaps of DEGs expressed by MEC, ductal and acini in DKO-M SMGs ($n = 5$) compared to WT-M ($n = 4$). The color scale represents scaled gene expression values. DEGs obtained using a non-parametric Wald test with Benjamini–Hochberg adjustment ($p$-value < 0.05 and fold change > 2). **c** qPCR confirmed gene expression changes in female and male SMGs normalized to *Rps29* and respective sex-matched WT (dotted line). WT-F and DKO-F: $n = 3$. WT-M and DKO-M: $n = 4$ or 7 SMGs. Unpaired two-tailed $t$-test compared to sex-matched WT (DKO-M: *Krt14* \*\*$p = 0.0045$, *Acta2* \*$p = 0.0188$, *Cnn1* \*\*$p = 0.0091$, *Myh11* \*\*$p = 0.0086$, *Mylk* \*$p = 0.0157$, *Tagln* \*$p = 0.0139$, *Postn* \*$p = 0.0152$, *Krt19* \*$p = 0.0127$, *Muc13* \*$p = 0.0206$, *Klk1b24* \*$p = 0.0183$, *Klk1b21* \*$p = 0.0382$, *Egf* \*$p = 0.0202$, *Ngf* \*$p = 0.0107$, *Fgf1* \*$p = 0.0159$, *Fgf9* \*$p = 0.0167$, *Pdgfa* \*$p = 0.0109$, *Tgfb2* \*$p = 0.0284$, *Mucl1* \*\*\*$p < 0.0001$, *Lpo* \*$p = 0.0338$, *Caecam1* \*\*$p = 0.0058$. DKO-F: *Krt14* \*$p = 0.0232$, *Acta2* \*$p = 0.0186$, *Cnn1* \*\*$p = 0.0010$, *Myh11* \*\*$p = 0.0082$, *Mylk* \*$p = 0.0321$, *Tagln* \*\*$p = 0.0026$, *Postn* \*$p = 0.0418$, *Klk1b24* \*$p = 0.0421$, *Klk1b21* \*$p = 0.0210$, *Ctgf* \*$p = 0.0191$, *Egf* \*$p = 0.0354$, *Ngf* \*$p = 0.0226$, *Pdgfa* \*$p = 0.0112$, *Kitl* \*$p = 0.0380$, *Aqp5* \*$p = 0.0163$, *Agt* \*$p = 0.0433$, *Mucl1* \*\*\*$p = 0.0004$, *Agr2* \*\*$p = 0.0092$). **d** Confocal imaging of WT-M and DKO-M SMGs showing Acinar1 (Ac1, green), Keratin-14 (K14, magenta), Mucin13 (M13, cyan), SMA (green) and EGF (magenta) immunostaining. Scale bar = 20 μm. **e** Quantification of staining normalized to the WT SMGs and nuclei. $n = 7$ (Ac1), 8 (K14, MUC13, EGF) and 9 (SMA) glands. Unpaired two-tailed $t$-test compared to WT. Ac1 \*\*\*$p < 0.0001$, K14 \*\*\*$p < 0.0001$, MUC13 \*\*\*$p < 0.0001$, SMA \*\*\*$p = 0.0001$, EGF \*\*\*$p = 0.0002$. All graphs show Mean ± SM. Source data are provided as a Source Data file.

Together, these data further suggest increased FGFR signaling in DKO acinar cells, which we predict may affect secretory unit function. In addition, increased MAPK signaling could also be activated via growth factor receptors other than FGFRs signaling.

## Calcium reabsorption, Ca²⁺-dependent volume change and apical-basal polarity are disrupted in DKO acinar secretory units

We next examined whether the secretory unit function was altered in the DKO SMGs. KEGG analysis showed calcium signaling and reabsorption, which are critical to secretion, were dysregulated in the DKO (Supplementary Fig. 2a). We measured intracellular $Ca^{2+}$ changes in isolated acinar cells following stimulation with carbachol (CCh), a muscarinic agonist that initiates signaling by releasing calcium ($Ca^{2+}$), which is detected with Fura2[35]. No differences were detected between WT and DKO cells in terms of CCh-induced $Ca^{2+}$ release as indicated by the first peak in the trace in $Ca^{2+}$ free external solution (Fig. 4c). However, a decrease in CCh-induced $Ca^{2+}$ influx in DKO acinar cells was detected when a $Ca^{2+}$ containing solution was injected (Fig. 4c). CCh-stimulated $Ca^{2+}$ entry in acinar cells is primarily mediated via the store-operated $Ca^{2+}$ entry pathway (SOCE), which is activated in response to depletion of ER $Ca^{2+}$ stores by hydrolysis, 1,4,5, trisphosphate ($IP_3$)[36]. This data suggests that the release of $Ca^{2+}$ from intracellular stores (endoplasmic reticulum) is normal in DKO, but the influx of extracellular $Ca^{2+}$ across the plasma membranes is disrupted in DKO acini. We also examined the effect of CCh stimulation on changes in cell volume as a readout of fluid secretion, using SMG lobule preparations, which contain both acinar and MECs. DKO acinar units had less CCh-induced cell volume loss (Fig. 4d, red trace in graph), and cell volume recovery was faster compared to WT, resulting in endpoint analysis showing a higher cell volume in DKO compared to WT (Fig. 4d, bar graph). Taken together, this suggests that fluid secretion was affected in the DKO. Secretion is highly dependent on acinar apical-basal polarity, so we stained acinar secretory units for tight junction protein-1 (ZO-1) and aquaporin 5 (AQP5), an apical water channel essential for saliva secretion. In accordance with the CCh stimulation data, apical ZO-1 and AQP5 staining in DKO were disrupted, showing that tight junction assembly and apical membrane polarity were affected (Fig. 4e), and thus salivary function may be affected.

## DKO mice have increased drinking behavior and reduced saliva flow

Basal salivary function was first assessed by using a Lickometer test which monitors drinking behavior by counting the number of licks mice take at the water port in 1 h. Female mice DKO mice had increased drinking behavior compared to their WT littermates, suggesting a reduction in basal salivary gland function (Fig. 4f). We then measured stimulated whole saliva flow after pilocarpine treatment of both male and female mice. Saliva flow was expressed as a ratio of saliva volume to body weight, the ratio was normalized to WT littermates, and there was an overall decrease in saliva flow in male DKO mice (~33% reduction) (Fig. 4g), whereas female DKO mice did not show a significant reduction (~17% reduction) (Supplementary Fig. 4c). The *Hs3st3a1* or *Hs3st3b1* single KO mice did not show any problems with saliva flow, although both these mice strains displayed increased drinking behavior.

Evaluation of stimulated saliva protein in the DKO saliva by gel electrophoresis and Coomassie Blue staining did not show any obvious differences or with the protein concentration of the saliva, although there was individual variability among mice (Supplementary Fig. 4b, c).

Taken together, our data suggest that in the adult DKO SMG, the loss of 3-*O*-HS domains disrupts the epithelial progenitor niche, increasing FGFR signaling and BM metabolism, disrupting MEC/acinar interactions, leading to reduced polarity and function. To identify the initial events causing this loss of homeostasis, we analyzed earlier developmental time points, predicting that increased FGFR signaling might increase BM synthesis by MECs and disrupt their development.

## Transcriptional changes during early SMG development suggest increased BM synthesis

In the DKO embryonic day (E) 13 SMG, when branching morphogenesis begins, there were no compensatory transcriptional changes in other 3-*O*-sulfotransferase enzymes (*Hs3st1, Hs3st2, Hs3st4, Hs3st5* and *Hs3st6*), although *Hs3st2, Hs3st4* and *Hs3st5* transcripts are barely detectable in the SMGs (Supplementary Fig. 5a). As expected, *Hs3st3a1* and *Hs3st3b1* were not detected while other HS biosynthetic genes did not change in DKO SMGs compared to WT (Supplementary Fig. 5). We analyzed genes known to be transcriptionally responsive to changes in FGFR signaling (*Fgfr1b, Fgfr2b, Etv4, Etv5*), as well as proliferation markers (*Ccnd1* and *Mki67*), BM components (*Col4a1, Col4a2, Hspg2, Lama3, Lama5, Lamb2, Lamb3*) and proteases involved in BM metabolism (*Mmp2, Mmp14* and *Mmp15*)[10,15,28,37]. There were no significant changes, although a trend of increased expression for *Col4a1, Col4a2, Hspg2* and *Mmp15* (>1.5 fold) suggesting changes in BM metabolism might be an early transcriptional readout for loss of 3-*O*-sulfation (Supplementary Fig. 5).

To determine potential changes in branching morphogenesis or gene expression, we performed ex vivo organ culture of E13 SMGs. We counted the endbuds at the beginning of culture and at 24 h showing a trend for increasing endbud number in the DKO (Fig. 5a). Gene expression analysis of the 3-*O*-sulfotransferase transcripts showed *Hs3st3a1* and *Hs3st3b1* were not detected and there was a decrease in *Hs3st2*, although expression is very weak. No other changes in gene expression were detected for any other HS biosynthetic genes analyzed (Fig. 5b). Changes in genes downstream of FGFR signaling (*Etv4* and *Etv5*) were not detected at this stage, there was increased laminin

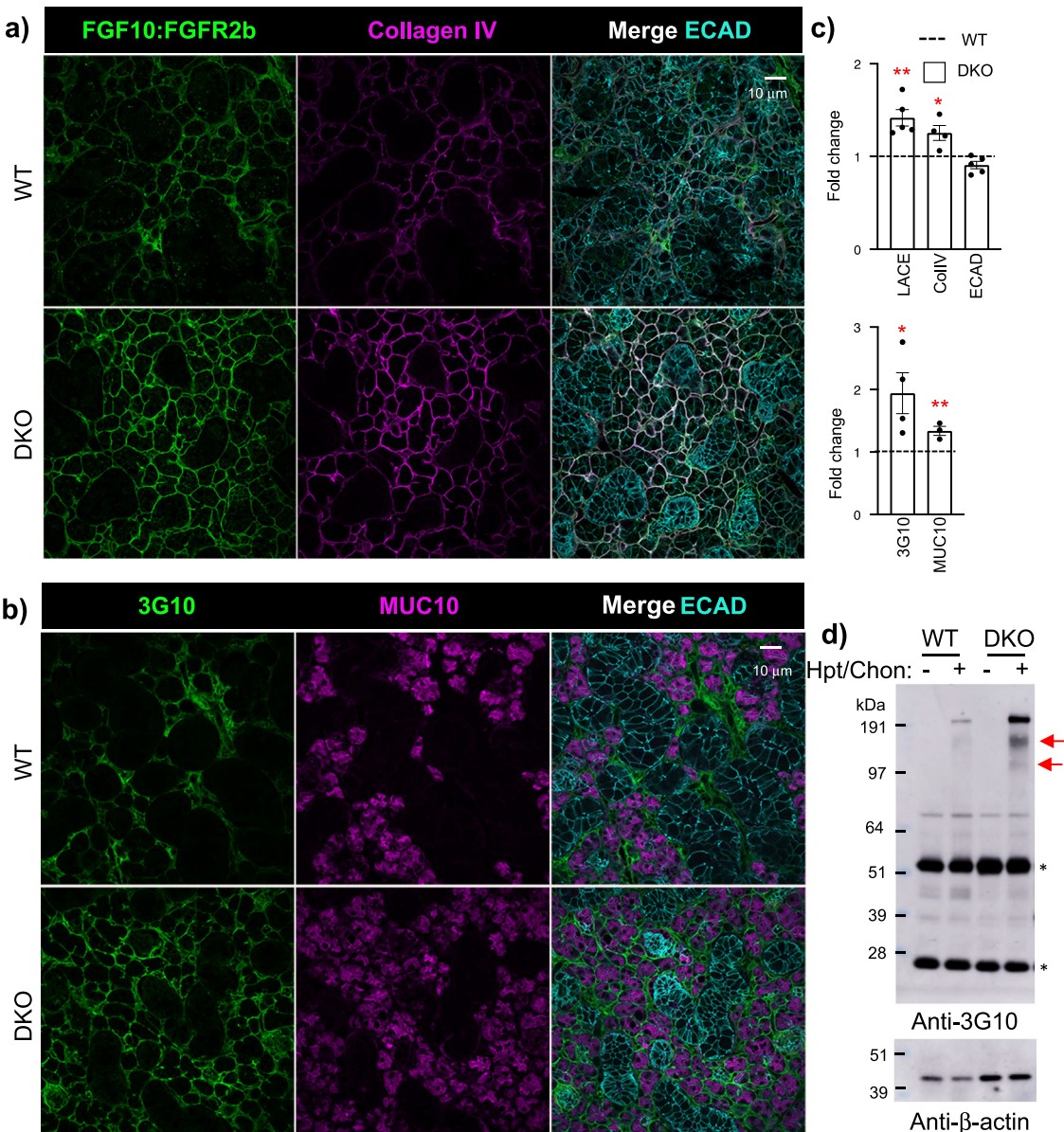

**Fig. 3 | DKO SMGs have increased binding of FGF10:FGFR2b-Fc complex (LACE) overlapping with the basement membrane and increased collagen IV, and HSPGs (3G10 staining) associated with acinar cells. a** Binding of FGF10:FGFR2b-Fc complex overlapping with COLIV, a basement membrane protein, is increased in DKO male SMGs. Representative images of single confocal sections from WT and DKO SMGs showing FGFR2b-Fc (green), COLIV (magenta), and E-cadherin (ECAD, cyan). Scale bar: 10 µm. **b** Representative images of single confocal sections from WT and *D*KO male SMGs showing 3G10, an antibody that binds HS stubs (green), acinar marker Mucin10 (magenta), and E-cadherin (ECAD, cyan). **c** Quantification of A and B protein fluorescence intensity normalized to total nuclei and expressed as a fold change compared to WT. LACE, ECAD: $n = 5$; COLIV, MUC10: $n = 3$; 3G10: $n = 3$ SMGs from WT and DKO mice. Error bars: SM. Unpaired two-tailed *t*-test compared to WT, LACE **$p = 0.0014$, COLIV *$p = 0.0431$, 3G10 *$p = 0.0102$, MUC10 **$p = 0.0063$. **d** Representative Western blot of 3G10 antibody, which detects the HS stub, shows an increase in high molecular weight HSPG cores (indicated by red arrows) in DKO male SMG lysates treated with or without heparinase III and chondroitinase ABC. β-actin is also shown. Bands identified by an asterisk may represent the light and heavy chains of immunoglobulins. $n = 6$ for WT and $n = 7$ for DKO SMGs. Source data are provided as a Source Data file.

gene expression (*Lamb2*, *Lamb3*) as well as a trend of increased expression of other BM genes and a proliferation marker (*Col4a1*, *Col18a*, *Hspg2*, *Ccnd1* all >1.5 fold) (Fig. 5b). These data suggest that loss of the highly 3-*O*-sulfated domain disrupts the BM, as HS chains of HSPGs on the cell surface and in the BM interact with both collagen IV and laminin networks. There was reduced immunostaining of the BM with HS4CV3 monoclonal antibody, which recognizes a 3-*O*-sulfated domain on HS, while agrin staining was not affected in E14 WT and DKO SMGs (Fig. 5c). In addition, staining with fluorophore-labeled antithrombin (AT488), which detects a 3-*O*-sulfated pentasaccharide (Hs3st1-like) in the BM, and perlecan, a BM HSPG (*Hspg2*) showed no

difference in E14 DKO SMGs (Fig. 5d). We predicted that the early transcriptional changes might have a functional consequence later in development when differentiation begins.

We therefore analyzed two later stages of development, at E15 near the onset of cell differentiation and postnatal day 2 (P2), where postnatal maturation and secretory function occur (Fig. 6a–d). MEC gene expression begins ~E15 and SMA (*Acta2*) immunostaining shows MECs appear in the outer layer of the endbud and differentiate and migrate to form stellate cells that wrap around the developing acinar structures from E17 to postnatal days P1-P8[28,38]. Mining previous scRNAseq of E16 epithelial cells shows BM genes are enriched in MECs

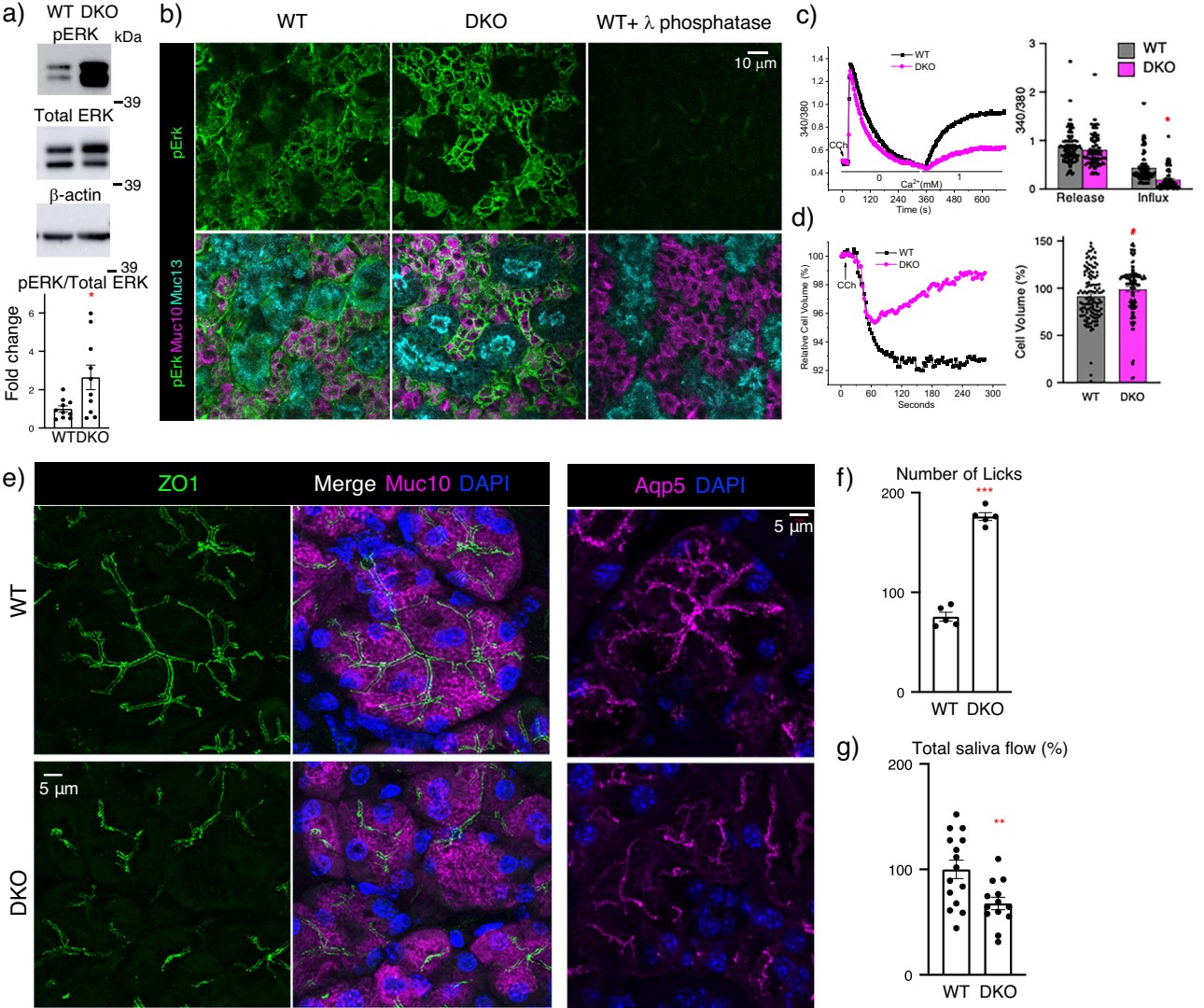

**Fig. 4 | Acinar MAPK, calcium signaling, and polarity are disrupted in DKO SMGs resulting in increased drinking behavior and reduced salivary secretion.**
**a** Western blots showing pERK is increased in DKO male SMG lysate. β-actin is also shown. Graph shows band intensity of pERK normalized to total ERK and expressed as a fold change in ratio compared to WT. The graphs are $n = 10$ biological samples from 3 experiments. Error bars: SM. Unpaired two-tailed $t$-test; *$p = 0.0309$.
**b** Immunostaining showing representative images of single confocal sections from WT and DKO male SMGs of pERK (green), the acinar Mucin10 (magenta), and ductal Mucin13 (cyan). Scale bar: 10 μm. White arrows point to MECs, and yellow asterisks identify ducts. $n = 9$ (WT) and $n = 10$ (DKO) SMGs. **c** Graphs show cytosolic Ca²⁺ change measurements stimulated by CCh with fura2 fluorescence (340/380). A representative trace from three independent experiments shows calcium reabsorption is decreased in DKO, and the graph shows the average of the group responses. Cells analyzed for calcium release studies are WT = 138 and DKO = 122

cells, and for calcium release WT = 136 and DKO = 122 cells. Error bars: SM. Unpaired two-tailed $t$-test; *$p = 0.0309$. **d** Recordings of CCh-regulated volume decrease (RVD) using calcein dye. Left-hand panel shows a representative trace and graph shows the average response of 125 WT and 125 DKO cells from three independent experiments. Error bars: SM. Unpaired two-tailed $t$-test compared to WT; *$p = 0.0343$. **e** Immunostaining for a polarity marker (ZO1), the acinar Mucin10 and AQP5. Nuclei are stained with DAPI (blue). Images are maximum projections of z-stacks from three biological samples. Scale, 5 μm. **f** Lickometer test of drinking behavior shows adult DKO female mice drink water more frequently than WT mice. Total number of licks compared to WT. Mean ± SM. Unpaired two-tailed $t$-test, ***$p < 0.0001$. $n = 5$ for both WT and DKO mice. **g** Salivary flow rates in adult male mice normalized to WT and shown as %. Mean ± SM. Unpaired two-tailed $t$-test, **$p = 0.0098$. $n = 15$ for WT and $n = 13$ for DKO mice. Source data are provided as a Source Data file.

(Supplementary Fig. 6a). Interestingly, qPCR analysis of E15 SMGs revealed upregulation of genes associated with FGFR signaling (*Fgfr1b, Fgfr2b, Etv4,* and *Etv5*), as well as MEC-expressed BM genes (*Col4a2, Col18a1, Col9a2, Lamb2, Lamb3, Hspg2*), and proteases (*Mmp14, Mmp15*) (Fig. 6a). There was also upregulated gene expression associated with MEC differentiation (*Ccn1, Mylk, Ntf5* and *Gpc3*). Neurotrophin gene expression increases MEC differentiation[28], and neurotrophin 5 (*Ntf5*) may drive MEC differentiation via Ntrk3. The qPCR analysis also showed that acinar cell markers (*Aqp5, Cldn10,* and *Lpo*) did not change. Immunostaining of E15 SMGs with SMA to label MECs, AQP5 to label acini, and BM perlecan (PLN) showed they were

similar in WT and DKO SMGs (Fig. 6b). Taken together, the loss of 3-O-HS leads to early transcriptional changes in BM homeostasis, leading to increased FGFR signaling that drives MEC differentiation and further BM synthesis, which may disrupt the progenitor microenvironment. However, after birth (P2), *Etv4* was the only gene related to FGFR signaling that was increased in the DKO, and there was reduced *Fgf9*, which is expressed in MECs. Many MEC genes, were similar to WT, but *Col18a1, Mylk, Myh11* and *Ngfb* were reduced, suggesting a reduction in MEC differentiation and secretory products (Fig. 6c). There was still increased *Mmp14* protease expression, and genes that mark ductal cells, such as *Klk1b21, Klk1b24* and *Krt5* were reduced, suggesting duct

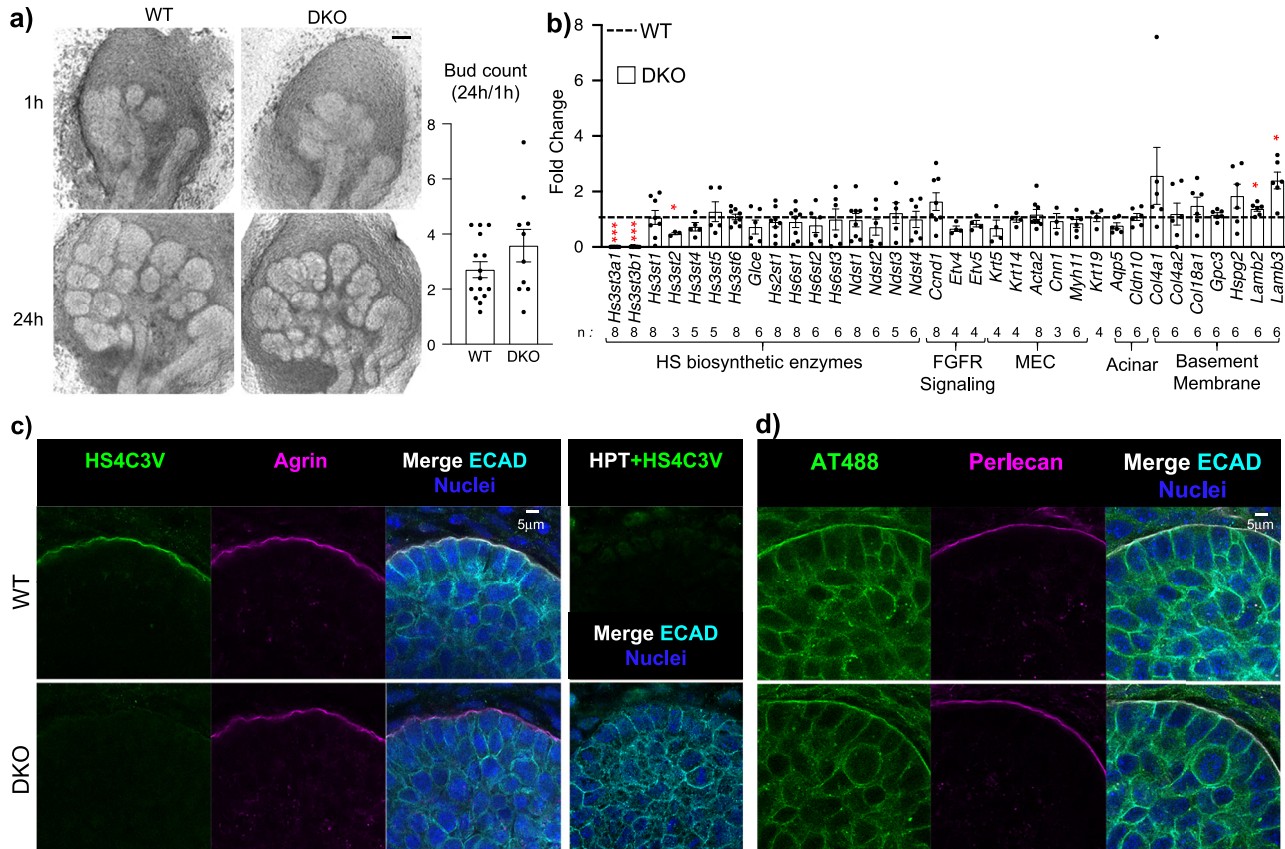

**Fig. 5 | Earlier in development, branching of E13 DKO SMGs in culture is not affected, although there is increased basement membrane gene expression.**
**a** Light micrograph of SMGs isolated from E13 embryos and cultured ex vivo for 24 h. Graph is quantification of the number of endbuds (expressed as a ratio of the number at 24 h/the number at 1 h) of WT (n = 15) and DKO (n = 10) SMGs. Error bars: SM. Scale bar, 100 μm. **b** Gene expression changes were normalized to WT and *Rps29*. Error bars: SM. The n values indicate the number of SMGs analyzed. Unpaired two-tailed *t*-test compared to WT (*Hs3st3a1* ***p < 0.001, *Hs3st3b1* ***p < 0.001, *Hs3st2* *p = 0.04, *Lamb2* *p = 0.0239 and *Lamb3* *p = 0.0103).
**c, d** Immunostaining at E13 shows Hs3st3-dependent 3-O-sulfation (HS4C3V) in the

basement membrane is reduced in DKO SMGs, whereas Hs3st1-dependent 3-O-sulfation (AT488 binding) is not affected. **c** Representative single confocal images of HS4C3V (green), agrin (magenta), E-cadherin (cyan) and nuclei in blue. Right panel shows that HS4C3V staining in the basement membrane is dependent on heparan sulfate. The section was pretreated with heparinase III enzyme (HPT), which removes the 3-O-sulfated epitope recognized by the antibody. Scale bars, 5 μm. n = 3 WT and n = 4 DKO embryonic glands. **d** Representative single confocal images of AT488 (green), perlecan (magenta), E-cadherin (cyan) and nuclei in blue. Scale bars, 5 μm. n = 3 WT and n = 3 DKO embryonic glands. Source data are provided as a Source Data file.

development was disrupted after birth. Overall, the data suggest that at the onset of postnatal development, MEC function is affected (Fig. 6c), reflecting the adult analysis (Fig. 2c). Importantly, immunostaining of MECs (SMA) and acinar cells (AQP5) showed a reduction in DKOs at P2, suggesting the disruption of secretory unit is evident by birth (Fig. 6d). These results led us to investigate whether growth and viability of DKO MECs were affected. We cultured P2 WT and DKO MECs on collagen IV-coated culture dishes as previously described[28]. After 7 days of culture, the DKO MECs appeared similar to WT, growing as large (>100 μm across) flat stellate cells (Supplementary Fig. 6b). However, the amount of total RNA extracted from DKO MECs was ~35% less than the WT MEC cultures, suggesting they grew less (Supplementary Fig. 6c), However, both protein and gene expression of DKO was similar to WT MECs. There were no differences observed in SMA, CNN1 or NGF staining (Supplementary Fig. 6c). In addition, no differences in genes related to FGFR signaling, MEC markers or MEC-enriched BM components were detected (Supplementary Fig. 6d). These data suggest that the progenitor potential of DKO MECs is not likely affected when they grow in similar collagen IV-containing microenvironment as WT. This also suggests that increased FGFR signaling in vivo affects the DKO MEC progenitor microenvironment.

## In vitro and ex vivo analyses of postnatal FGFR function
Finally, to determine how changes in levels of Hs3st1- or Hs3st3-type 3-O-sulfation could directly increase FGFR signaling or MEC microenvironment, we took advantage of a recent advance in HS biosynthesis that has enabled the chemoenzymatic synthesis of HS 12mers with defined 3-O-sulfation and biotin attached. The three 12mer HS all have 2-O, 6-O and N-sulfation and one has one Hs3st1-type 3-O-sulfate (3ST1) or a Hs3st3-type 3-O-sulfate (3ST3) (Fig. 7a). We used streptavidin bead pulldown assays to compare complex binding with recombinant FGF1, FGF2, FGF7 and FGF10 with FGFR1b and FGFR2b (Fig. 7b). In these assays we confirmed that all three of these 12mer HS were able to pull down FGFR/FGF complexes, and so could be biologically active to compete with endogenous HS functions in ex vivo assays or to enhance signaling complex formation (Fig. 7b, c). Interestingly, our data suggest that Hs3st1-HS has increased complex formation with both FGFR1b and FGFR2b complexed with either FGF2 or FGF10 and less complex formation with FGFR2b. As expected, FGFR1b does not bind FGF7, or to the beads alone, which were used as negative controls (Fig. 7b). Once we confirmed that these 12mers could form FGF/FGFR ternary complexes that are required for signaling, we added them to FGF10-dependent epithelial assays[15] and measured epithelial morphogenesis, gene expression related to FGFR signaling and MEC

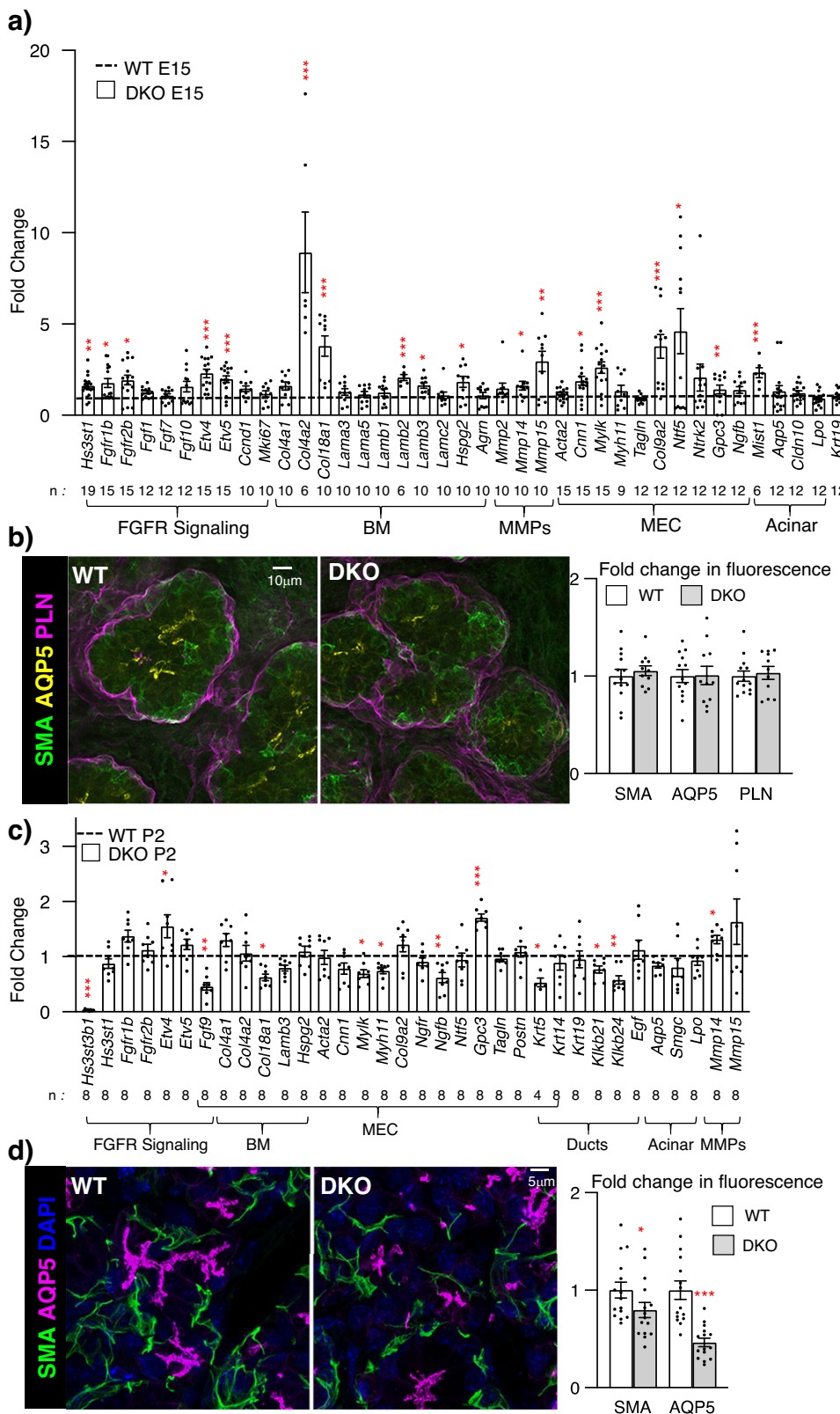

differentiation. Both 3-O-sulfated 12mers stimulated more epithelial morphogenesis than the control HS, although the 3ST3 had the greatest effect on morphogenesis (Fig. 7c). Surprisingly after 48 h both 3-O-sulfated 12mers stimulated more FGFR genes (*Etv4*), MEC specific gene expression (*Acta2* and *Cnn1*), acinar gene (*Aqp5*), *Hs3st3a1*, and BM HSPG (*Col18a1*). A reduction in ductal gene (*Krt19*), *Hspg2* and *Lamb3* was detected in epithelia treated with 3ST3-HS (Fig. 7d).

Immunostaining for MEC markers SMA and CNN1 (Fig. 7e) showed that 3ST3 stimulated more MEC differentiation, and the entire outer epithelial layer expressed SMA and individual SMA+ cells started to express CNN1 (Fig. 7e). Taken together, the 12mer HS increases both FGFR1b and FGFR2b complexes to increase FGFR signaling which stimulated MEC differentiation and production of BM components. In this assay, the expansion of the epithelial endbuds resulted in a

**Fig. 6 | E15 DKO SMGs have increased gene expression downstream of FGFR signaling, basement membrane components and MEC markers, but by P2 MEC gene expression is reduced. a** Gene expression changes in E15 DKO SMGs were normalized to WT control and *Rps29*. Error bars: SM. The n values indicate the number of SMGs analyzed. Unpaired two-tailed *t*-test compared to WT (*Hs3st1* **$p = 0.0014$, *Fgfr1b* *$p = 0.0167$, *Fgfr2b* *$p = 0.0229$, *Etv4* ***$p < 0.0001$, *Etv5* ***$p = 0.001$, *Col4a2* ***$p = 0.0002$, *Col18a1* ***$p = 0.0001$, *Lamb2* ***$p = 0.0004$, *Lamb3* *$p = 0.0119$, *Hspg2* *$p = 0.0418$, *Mmp14* *$p = 0.0191$, *Mmp15* **$p = 0.009$, *Cnn1* *$p = 0.0316$, *Mylk* *$p = 0.0004$, *Col9a2* ***$p = 0.0002$, *Ntf5* *$p = 0.0441$, *Gpc3* *$p = 0.0086$, *Mist1* ***$p = 0.0007$). **b** Representative images show immunostaining at E16 with SMA (green) in MEC, AQP5 (yellow) in acinar apical membranes, and perlecan (magenta) in basement membrane. Scale bar, 10 μm. Graph shows quantification of protein fluorescence intensity normalized to total nuclei and expressed as a fold change compared to WT. $n = 13$ WT and 11 DKO SMGs. Error bars: SM. **c** Gene expression changes in P2 DKO SMGs were normalized to WT control and *Rps29*. Error bars: SM. The n values indicate the number of SMGs analyzed. Unpaired two-tailed *t*-test compared to WT (*Hs3st3b1* ***$p < 0.0001$, *Fgf9* **$p = 0.0029$, *Gpc3* ***$p < 0.0003$, *Klk1b21* *$p = 0.0201$, *Klk1b24* **$p = 0.0036$, and *Mmp14* *$p = 0.0378$). **d** Representative images showing immunostaining at P2 show SMA (green) in MEC, AQP5 (magenta) in acinar apical membranes, and nuclei (DAPI, blue). Scale bar, 5 μm. Graph shows protein fluorescence intensity quantification normalized to total nuclei and expressed as a fold change compared to WT. $n = 15$ WT and 15 DKO SMGs. Error bars: SM. Unpaired two-tailed *t*-test compared to WT, SMA *$p = 0.0475$, AQP5 ***$p < 0.0001$. Source data are provided as a Source Data file.

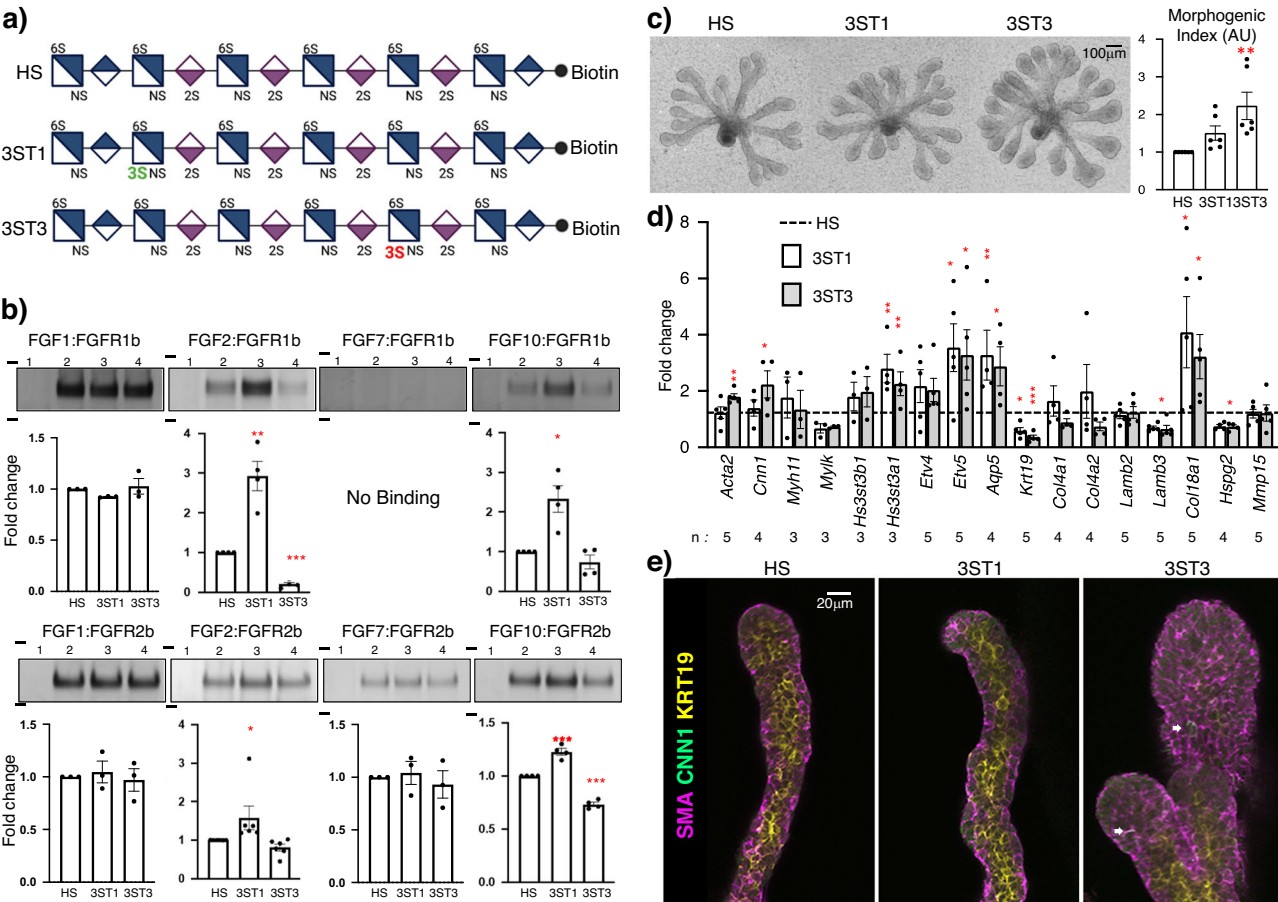

**Fig. 7 | Chemoenzymatically synthesized 12mer 3-*O*-HS forms complexes with FGFs and both FGFR1b and rFGFR2b and increases MEC differentiation in epithelia culture. a** Chemical structure of chemoenzymatically synthesized 12mer HS with biotin tag. Figure 7a Created with BioRender.com released under a Creative Commons Attribution-NonCommercial-NoDerivs 4.0 International license. **b** Pulldown assays of 12mer HS and FGF:FGFR complexes. Top panel show representative gels for FGFs 1,2,7 and 10 complexed with rFGFR1b, bottom panel shows the same FGFs with rFGFR2b. Lane 1: beads only, lane 2: non-3-*O*-HS, lane 3: 3ST1-HS, lane 4: 3ST3-HS. The molecular weight markers in kilodaltons (97 and 64 kDa) are indicated on each gel. Full gels are provided in the source data file. Graphs show quantitation of the FGFR complex separated by SDS-PAGE and silver-stained from independent samples. $n = 3$ for FGF1:FGFR1b, FGF1:FGFR2b, FGF7:FGFR1b, FGF7:FGFR2b; $n = 4$ for FGF2:FGFR1b, FGF10:FGFR1b, FGF10:FGFR2b; and $n = 6$ for FGF2:FGFR2b. Error bars: SM. One-way ANOVA with Dunnett's test for multiple comparisons to non-3-*O*-HS, FGF2:FGFR1b: 3ST1-HS **$p = 0.0026$, 3ST3-HS ***$p < 0.0001$; FGF10:FGFR1b: 3ST1-HS *$p = 0.0136$; FGF2:FGFR2b: 3ST1-HS *$p = 0.0482$; FGF10:FGFR2b: 3ST1-HS ***$p = 0.0002$ and 3ST3-HS ***$p < 0.0001$. **c** E13 SMG epithelia cultured with FGF10 in a laminin-111 3D matrix were treated with 12mer HS and cultured for 48 h ($n = 6$ with 4 epithelia each), the size of the epithelia was measured as a morphogenic index. Error bars: SM. One-way ANOVA with Dunnett's test for multiple comparisons to non-3-*O*-HS, 3ST3-HS ***$p = 0.0006$. **d** Gene expression of markers of FGFR signaling and MECs were analyzed by qPCR were normalized to non-3-*O*-HS and *Rps29*. Error bars: SM. The n values indicate the number of epithelial culture experiments analyzed. One-way ANOVA with Dunnett's test for multiple comparisons to non-3-*O*-HS (3ST1-HS: *Hs3st3a1* **$p = 0.0017$, *Etv5* *$p = 0.0129$, *Aqp5* **$p = 0.0070$, *Krt19* *$p = 0.0317$, *Col18a1* *$p = 0.0118$ and for 3ST3-HS: *Acta2* **$p = 0.0078$, *Cnn1* *$p = 0.0496$, *Hs3st3a1* **$p = 0.0084$, *Etv5* *$p = 0.0181$, *Aqp5* *$p = 0.00148$, *Krt19* ***$p = 0.0004$, *Lamb3* *$p = 0.0128$, *Col18a1* *$p = 0.0211$, *Hspg2* *$p = 0.0194$). **e** Confocal imaging of E13 epithelia treated with 12mer HS showing SMA (magenta) and CNN1 (green) and KRT19 (yellow). White arrows point to CNN1 cells. Scale bar, 20 μm. $n = 3$ independent epithelia experiments. Source data are provided as a Source Data file.

proportional reduction in duct development. These assays highlight the cell-type specificity and microenvironmental complexity of 3-O-HS regulation of FGFR signaling and MEC differentiation. We propose a model where 3ST1-HS and 3ST3-HS can both promote FGFR complex formation and signaling in specific cellular contexts during development. We begin to tease apart this complexity by reducing 3ST3-HS in vivo, showing that manipulating the balance of types of 3-O-sulfation in vivo affects the levels of FGFR signaling and BM metabolism and has consequences for the intercellular communication during development. Increased FGFR signaling and BM synthesis have a detrimental effect on MEC progenitor microenvironment, reducing MEC growth and function postnatally, which reduces the function of the acinar secretory unit.

## Discussion

Here, we knocked out two *Hs3st3* enzymes to investigate 3-O-HS regulation of FGFR signaling and MEC function during SMG development and homeostasis. We confirmed the loss of specific 3ST3-HS tetrasaccharides in the SMGs. DKO mice were smaller, suggesting overall reduced growth factor function during development, but their SMGs were proportionally larger with a disrupted epithelial compartment. The reduction of MEC and ducts suggested an altered progenitor microenvironment. We show that increased BM gene expression was an early event that we propose led to further increases in genes related to FGFR signaling and MEC differentiation. However, this dysregulation of the progenitor microenvironment disrupted MEC and acinar cell communication reducing secretory unit function. Furthermore, we show that both types of 3-O-HS can form multiple FGF/FGFR signaling complexes, and in an FGF10-dependent ex vivo assay, 3ST3-HS increased MEC differentiation.

Although the present work focuses on the *Hs3st3a1*[−/−];*Hs3st3b1*[−/−] DKO, it was apparent that 3ST1-HS is important for FGFR signaling (Fig. 7b), therefore we also crossed *Hst3st1*[−/−] mice[30] with the single *Hs3st3b1*[−/−] mice, as these are the two most abundant isoforms in SMGs[15]. Loss of *Hs3st1* alone is not lethal[31] however, we were unable to generate live *Hs3st1*[−/−];*Hs3st3b1*[−/−] pups at weaning and further analysis showed they die by E11 (Supplementary Fig. 7a). We obtained 3 DKO embryos at E11 by crossing an *Hs3st1*[−/−];*Hs3st3b1*[+/−] with a *Hs3st1*[+/−];*Hs3st3b1*[+/−]. Gross histological analysis shows that loss of both enzymes results in early cardiac developmental defects that likely cause the lethality, highlighting that 3-O-HS is essential for embryonic development (Supplementary Fig. 7b). Further study will require a generation of conditional knockouts to investigate the function of both types of 3-O-HS later in development.

Here we focused on MECs, which are located between acinar and duct cells and their surrounding BM. MECs are a communication hub providing regulatory signals between the epithelial and stromal compartments during development[28]. Mammary luminal cells cultured in 3D collagen without MECs form acinar-like structures that are not polarized and lack lumens, however inclusion of MECs rectifies this[26]. In our analysis, DKO secretory units had disrupted acinar apical-basal polarity, a likely consequence of improper MEC communication and/ or BM metabolism. Although there was reduced transcription of some BM genes enriched in MECs (Supplementary File 1), the protein staining of collagen IV and HSPGs around acinar cells was increased, suggesting BM metabolism was affected. It is not clear if the increased BM staining is related to a lack of proteolytic turnover of BM components, which needs further investigation.

The protein interactome with 3-O-HS domains includes antithrombin-III, glycoprotein D of HSV-1, neuropilin-1, cyclophilinB, stabilin, Serpin D1, FGFR1, FGF7, FGF9, RAGE, BMP2[13,30,39–45]. Here, the reduction of 3ST3-HS altered specific FGF/FGFR complex binding to the BM HS surrounding acinar cells, raising the issue of how the balance of the 3-O-sulfated domains regulates FGFR functions. We speculate that the Hs3st3-sulfated HS domains may restrict or limit FGF diffusion and/or

FGF/FGFR interaction in the BM, and our pulldown assays suggest Hs3st1-derived domains have increased binding to multiple FGF/FGFR complexes. Previous reports[39] using glycan arrays showed that FGF7 and FGFR1b directly interact with a pentasaccharide with similar sulfation as the tetrasaccharide that is not detected in the DKO. We speculate that FGF7 is more freely diffusible through the BM to stimulate FGFR2b signaling. Postnatal MECs produce FGF7[1], but at E15, the mesenchyme is still the major source of FGF7, which would have to diffuse through the BM, which supports the concept that 3-O-HS has different roles in different cells at discrete stages of development.

HS modulates the dynamics and kinetics of ligand-receptor interactions, which can influence the duration and potency of signaling. In the adult DKO, the increased signaling in acinar FGFR2b results in increased pERK staining. It is well known that pERK is a key downstream component of the RAS/MEK/ERK signaling and can be translocated to the nucleus, where it regulates gene expression by phosphorylating transcription factors[46,47]. Nuclear ERK activity mediates proliferation, whereas ERK-dependent differentiation is associated with ERK activity in the cytoplasm. We recently reported that FGFR2b was essential for seromucous acinar differentiation[1]. In line with this, the increase in seromucous acinar cells observed in adult DKO further supports that FGFR-dependent signaling increases with the loss of the 3-O-HS. In addition, our data suggest that FGF2 and FGF10 signaling via FGFR1b may also play a role in acinar development. While we focused on FGF10 signaling in isolated epithelial assays, acinar differentiation is also driven by FGF2 signaling[48] and the Nrg1/ Erbb3 pathway[49], which was not transcriptionally affected in the DKO.

In the adult DKO, pERK was localized to the acinar cytoplasm and associated with disrupted polarity (Fig. 4e), which is similar to the lacrimal gland, where MECs support acinar polarity and structure[50]. We speculate the loss of 3-O-HS disrupts BM metabolism allowing increased FGF diffusion and increasing acinar FGFR signaling. The identity of BM or cell surface HSPGs that are responsible for this remains to be determined. We also speculate that the disrupted acinar polarity and the lack of contractile MEC to expel saliva leads to the increase in gland weight due to saliva production by acinar secretory units that cannot secrete.

In summary, we propose a model where the genetic deletion of *Hs3st3a1* and *Hs3st3b1* results in the loss of the 3-O-sulfated domains in the BM and the cell surface, increasing growth factor signaling to MECs. This promotes gene expression associated with BM metabolism, potentially generating more bioactive proteolytic BM products, which increase FGFR signaling and disrupts MEC function. MECs produce more BM proteins and HSPGs, likely altering BM architecture and MEC-acinar communication, which disrupts acinar polarity and impairs calcium signaling, affecting acinar secretory unit function. Understanding how the 3-O-sulfated HS code regulates FGFR signaling, BM metabolism and MEC biology will be essential to manipulate cellular specificity of growth factor signaling and enhance the progenitor microenvironment to drive organ regeneration.

## Methods

### Materials availability

No specific new materials were generated in this work, see Supplementary Table 2 for resources.

### Mouse strains

*Hs3st3a1* and *Hs3st3b1* knockout mice were generated using ES cells obtained from the KOMP repository as previously described[17]. *Hs3st1* KO mice[30,31] were obtained from Dr. Jeff Esko (University of California, San Diego) crossed in a C57BL6 background. The generation of the *Hs3st3a1* and *Hs3st3b1* double-knockout (DKO) mouse has previously been described[19]. Mice were kept in a 14 h on/10 h off light/dark cycle at 74–78 F with 50–70% humidity. To generate DKO embryos used in the study, timed mating was set up and a plug detected was considered

day 0. When using adult mice for experiments, due to the known sexual dimorphism in adult mouse SMGs, differing outcomes between sexes was considered. Sex was not considered in experiments using embryonic or neonatal glands, as sexual dimorphism does not occur during development. All mice were maintained and treated according to guidelines approved by the National Institute of Dental and Craniofacial Research and the National Institutes of Health Animal Care and Use Committee.

## Genotyping

Mice were genotyped for *Hs3st3b1* allele as previously described[17]. For *Hs3st3a1* amplification, *Hs3st3a1* ZFN primer Fwd (CTGGCCTTA CTTCTGGACGA) and Hs3st3a1 ZFN Rev (CAAGGGAGAAGAACGGGAG) were used for the amplification of DNA. PCR cycling parameters included an initial denaturing at 94 °C for 5 min; denaturing at 94 °C for 15 s, annealing at 55 °C for 30 s, extension at 72 °C for 40 s, repeated for thirty cycles. Final extension: 72 °C for 5 min. To perform the mismatch-sensitive nuclease assay, PCR products were denatured with T7E1 enzyme (New England BioLabs, Ipswich, MA, USA) at 95 °C for 2 min; followed by a 2 °C/s drop to 85 °C, and a further 0.1 °C/s drop to 25 °C. PCR products were separated on a 2.0% TBE agarose gel with ethidium bromide. To identify mismatches, PCR products were purified using QIAquick PCR purification kit (QIAGEN Inc., Valencia, CA, USA) and sequenced.

## Extraction and quantitation of HS from SMGs

The procedure for isolating HS from SMGs is published with minor modifications[51]. Briefly, 3–4 SMGs were pooled together for each sample, homogenized, and defatted in chloroform and methanol. The defatted tissues were dried and weighed to obtain the dry weight. The dried tissue was subjected to proteolysis with pronase E (10 mg:1 g (w/w), pronase E/protein) at 55 °C for 24 h. The solution was then denatured and centrifuged to obtain a supernatant. Prior to subjecting to DEAE column purification, the recovery calibrant $^{13}$C-labeled *N*-sulfo heparosan was added to the supernatant. DEAE column mobile phase A consisted of 20 mM Tris, pH 7.5 and 50 mM NaCl, and mobile phase B consisted of 20 mM Tris, pH 7.5 and 1 M NaCl. The column was washed with 1.5 mL mobile phase A, and then HS fraction was eluted with 1.5 mL mobile phase B. The eluted was desalted using the YM-3KDa spin column, and the retentate was subjected to heparin lyase digestion. The 100 µL of digestion solution (7.5 µL of enzymatic buffer (100 mM sodium acetate/2 mM calcium acetate buffer (pH 7.0) containing 0.1 g/L BSA), and 1.25 µL heparin lyases I (2.49 mg/mL) and 2.5 µL heparin lyases II (13.6 mg/mL)) was incubated at 37 °C for 5 h. Prior to recovering the digests, a known amount of $^{13}$C-labeled non-3-*O*-sulfated disaccharide calibrants were added to the digestion solution[18]. The HS disaccharides and tetrasaccharides were recovered by centrifugation. The collected filtrates were freeze-dried before the 2-Aminoacridone (AMAC) derivatization. A total of five $^{13}$C-labeled 3-*O*-sulfated oligosaccharide calibrants were used during the analysis of 3-*O*-sulfated tetrasaccharides. These structures are oligo 1 GlcNAc-GlcA-GlcNAc6S-GlcA*-GlcNS3S6S-IdoA2S*-GlcNS6S-GlcA-pNP, oligo 2 GlcNAc-GlcA-GlcNS6S-GlcA*-GlcNS3S6S-IdoA2S-GlcNS6S-GlcA-pNP, oligo 3 GlcNS6S-GlcA*-GlcNS6S-IdoA2S*-GlcNS3S6S-IdoA2S*-GlcNS6S-GlcA-pNP, oligo 4 GlcNS-GlcA-GlcNS-IdoA2S*-GlcNS3S-IdoA2S-GlcNS-GlcA-pNP and oligo 5 GlcNAc-GlcA-GlcNS-IdoA2S*-GlcNS-IdoA2S-GlcNS3S-IdoA2S-GlcNS-GlcA-pNP.

Eight $^{13}$C-labeled HS disaccharide calibrants were used for the analysis of the non-3-*O*-sulfated HS portion, and their structures were published[51].

## Chemical derivatization of HS disaccharides and tetrasaccharides

The AMAC derivatization of lyophilized samples was performed as described previously[51]. Briefly, 10 µL of 0.1 M AMAC solution in DMSO/glacial acetic acid (17:3, v/v) was added to the samples and incubated at room temperature for 15 min. Then 10 µL of freshly prepared 1 M aqueous sodium cyanoborohydride was added to this solution. The reaction mixture was incubated at 45 °C for 2 h. After incubation, the reaction solution was centrifuged to obtain the supernatant, which was subjected to the LC-MS/MS analysis.

## LC-MS/MS analysis

The analysis of AMAC-labeled HS was performed as previously described with slight modifications[51,52]. A Vanquish Flex UHPLC System (ThermoFisher Scientific) coupled with TSQ Fortis triple-quadrupole mass spectrometry as the detector was used. The ACQUITY Glycan BEH Amide column (1.7 µm, 2.1 × 150 mm; Waters, Ireland, UK) was used to separate di/tetra-saccharides at 60 °C. Mobile phase A buffer used was 50 mM ammonium formate in water, pH 4.4, whereas Mobile phase B was acetonitrile. The flow rate of 0.3 mL/min and the elution gradient as follows were used: 0–15 min 83–70% B, 15–30 min 70–50% B, 30–35 min 50% B, 35–45 min 83% B. The. Online triple-quadrupole mass spectrometry operating in the multiple reaction monitoring (MRM) mode was used as the detector. The ESI-MS analysis was operated in the negative ion mode using the following parameters: Negative ion spray voltage at 3.0 kV, sheath gas at 55 Arb, aux gas at 25 Arb, ion transfer tube temp at 250 °C and vaporizer temp at 400 °C. TraceFinder software was applied for data processing. The amount of HS was determined by comparing the peak area of native di/tetra-saccharides to each di/tetra-saccharides calibrant, and the recovery yield was calculated based on the comparison of the amount of recovery calibrant disaccharide in the samples and control, respectively. Due to batch-batch variability, HS analysis of DKO and single KO males or WT and DKO male and female SMGs was performed on samples analyzed within the same analytical runs.

## RNA-sequencing analysis

For RNAseq, 3 SMGs each were used from WT and DKO female mice, whereas 4 WT and 5 DKO SMGs were used from male mice to isolate RNA. All RNAseq analysis for the SMG cDNA libraries was performed at the NIDCD/NIDCR Genomics and Computational Biology Core using the Nextera XT method as previously described[53]. DeSeq2 was used to perform statistical test analysis to assess differentially expression genes (DEG) between groups with a log2 fold in expression with the adjusted *p*-value of <0.05, as previously described in ref. 54. RNA-seq data are available on the Gene Expression Omnibus (GEO) website (GEO: GSE235187). DEGs were also used to identify the enriched Kyoto Encyclopedia of Genes and Genome (KEGG) pathways [16].

## Evaluation of scRNAseq gene expression

Ready-to-use .rds objects from publicly available scRNAseq data of embryonic SMGs were downloaded (GSE150327, https://sgmap.nidcr.nih.gov)[24]. Scripting was done in R and the SEURAT package[55,56] was used for visualization of gene expression. The E16 timepoint was extracted from the integrated SEURAT objects using the SplitObject function from SEURAT and DotPlots were generated with the DotPlot function of the same package.

## Quantitative PCR (qPCR)

Real-time PCR was performed as previously described[17]. Briefly, murine SMGs were minced and lyzed by homogenization. RNA was isolated using the RNAqueous-4PCR total RNA isolation kit and DNAse reagents according to the manufacturer's instructions (ThermoFisher Scientific). cDNA was generated from DNase-free RNA samples, amplified and expression was normalized to the housekeeping gene, *Rps*29. The reactions were run in duplicates, and with at least three biological samples. See Supplementary Table 2 for specific primer sequences, which were designed using Beacon Desgner™ software (PREMIER Biosoft).

## Histology

SMG tissue from adult male and female WT and DKO mice were isolated and fixed in 4% PFA before standard paraffin embedding was performed by Histoserv Inc (Germantown, MD). Hematoxylin and Eosin (H&E) staining was performed on 5 μm thick sections (Histoserv Inc, Germantown, MD). Slides were scanned using ×40 objective with either a S60 Nanozoomer Digital slide scanner (Hamamatsu) or Aperio Scanscope scanner (Leica Biosystems), and snapshots of random areas (250 μm²) were taken using similar settings in either NDP.view 2 software (Hamamatsu) or Aperio Image software. Blinded quantification of ducts and acinar cells per area was performed by using the cell counter setting for manual counting in FIJI. Between 13 and 21 areas from 3 to 4 glands/animals were used for quantification (Male WT: $n = 21$, male DKO: $n = 20$, female WT: $n = 19$, female DKO: $n = 12$).

## Immunofluorescence staining

Paraffin-embedded sections were analyzed with primary antibodies specific for SMA (Millipore Sigma, St Louis, MO, USA. #A2547), AQP5 (Alomone, Jerusalem, Israel. #AQP-005), e-cadherin (Cell Signaling Technology, Boston, MA, USA. #3195), e-cadherin (BD Biosciences, San Jose, CA, USA. #610182), KRT14 (Covance, #PRB-155P), Mucin13 (Santa Cruz Biotechnology, sc-390115), acinar 1 (DBHB, 3.7A12), EGF (gift from Dr. Edward W. Gresik, CUNY), collagen type IV (EMD Millipore, #AB756P (rabbit)), #AB769 (goat), anti-delta-heparan sulfate 3G10 (asmbio, #370260-1), Mucin10/Prol1 (Everest, #EB10617), p44/p42 (ERK1/2) (Cell Signaling, 137F5 #4695), phospho-p44/p42 MAPK (ERK1/2) (Thr202/Tyr204) (Cell Signaling, D13.14.4E #4370), tight junction protein-1 ZO-1 (Invitrogen, # 339100), CNN1 (Abcam, ab46794), Refer to Supplementary Table 2 for antibody details.

Fluorescence immunostaining on paraffin sections of P2 and adult SMGs was performed as previously described[17]. Briefly, following de-paraffinization in xylene substitute and re-hydration in graded ethanol solutions, tissue sections were subjected to heat-mediated antigen retrieval using R-universal epitope recovery buffer (Electron Microscopy Sciences, Hatfield, PA, USA) or Tris-EDTA pH 9 buffer for 10 min at high-pressure. Sections were blocked for 1 h at room temperature with 10% heat-inactivated donkey serum (Jackson ImmunoResearch Laboratories, Westgrove, PA, USA), 1% BSA (Sigma, St Louis, MO, USA) and Mouse on Mouse (MOM) blocking reagent (Vector Laboratories, Burlingame, CA, USA). The slides were incubated overnight at 4 °C with the appropriate primary antibody, then washed with PBS containing 0.1% Tween-20 (PBST) and incubated with the respective dye-conjugated secondary antibodies (all from Jackson ImmunoResearch Laboratories, Westgrove, PA, USA) and DAPI stain (EMD Millipore Corp, Billerica, MA, USA) for 1 h at room temperature. Following washes with PBST, slides were mounted with Fluoro-Gel mounting medium (Electron Microscopy Sciences, Hatfield, PA, USA). Phospho-ERK and 3G10 staining were performed as previously described[53].

## Whole-mount immunostaining

Whole-mount immunostaining on E13 SMGs was performed as previously described[17,15]. Antibodies used for whole-mount immunostaining include E-cadherin (BD Biosciences, San Jose, CA, USA. #610182), agrin (R &D, #AF550), perlecan (Millipore Sigma, #mAb1948), HS4C3V single-chain HS Ab (Dr. Toin H. van Kuppevelt (Radboud University Medical Center, The Netherlands)), and AT488 (gift from Dr. Nicholas Shworak (George Washington University)). Refer to Supplementary Table 2 for antibody details.

Briefly, SMGs were fixed with 4% PFA in PBS and permeabilized with 0.1% triton-X-100. SMGs were blocked for 1 h at room temperature with 10% heat-inactivated donkey serum (Jackson ImmunoResearch Laboratories, Westgrove, PA, USA), 1% BSA (Sigma, St Louis, MO, USA) and MOM blocking reagent (Vector Laboratories, Burlingame, CA, USA). Then incubated overnight at 4 °C with the appropriate primary antibody, then washed with PBST and incubated with the respective dye-conjugated secondary antibodies (all from Jackson ImmunoResearch Laboratories, Westgrove, PA, USA) and DAPI stain (EMD Millipore Corp, Billerica, MA, USA) for 1 h at room temperature. Following washes with PBST, SMGs were mounted with Fluoro-Gel mounting medium (Electron Microscopy Sciences, Hatfield, PA, USA). Staining for anti-HS single-chain HS4C3V antibody (1:50) was performed as previously described[17]. Staining using AT488 was performed as previously described[15].

Whole-mount staining on E16 SMGs was performed by permeabilizing the SMGs in 0.2% Triton-X-100/PBS at 4 °C for 48 h, followed by blocking as described above. The glands were incubated with the primary and secondary antibodies at 4 °C for 48 h. All slides were imaged using a Zeiss 880 confocal microscope, and, in some cases, a super-resolution airy scan was used for imaging. Fluorescence was quantified in the epithelial region within the basement membrane (ROI) and normalized to that of nuclei in the ROI, and then the data was normalized to control. An average of five images were taken per SMG, and a minimum of three glands were used for each group.

For whole-mount staining of isolated epithelia, epithelia were fixed with ice-cold acetone-methanol for 10 min at −20 °C. The epithelia were rinsed with PBST, and antibody incubations were performed as described for the whole-mount staining of E13 SMGs.

## Ligand and carbohydrate engagement (LACE) assay

A modification of the LACE assay[57] was performed using a recombinant mouse FGFR2b-human Fc chimera (referred to as rFGFR2b), FGFR1b-human Fc (rFGFR1b), FGF1, FGF7 and FGF10 (all from R&D Systems Inc., MN) as previously described in refs. 17,53. Briefly, adult glands fixed with 4% paraformaldehyde overnight at RT were processed for paraffin sectioning. Heat-activated antigen retrieval using a 0.05% Tween-20/10 mM citric acid buffer pH 6.0 was performed on 7 μm sections. The samples were blocked for 1 h at RT with 10% heat-inactivated donkey serum, 1% BSA, and 1.8% MOM IgG blocking reagent, incubated overnight at 4 °C with 50 nM soluble recombinant rFGFR2b-Fc, 50 nM recombinant FGF10, anti-E-cadherin (BD Transduction Labs) and anti-collagen TIV (Millipore). After washing, Cy dye-conjugated secondary antibodies were added for 1 h. The Fc on the receptors was detected with Alexa Fluor® 488 AffiniPure F(ab′)$_2$ Fragment Donkey Anti-Human IgG, Fcγ fragment specific secondary antibody (Jackson ImmunoResearch Laboratories, Inc). Immunofluorescence was analyzed with a Zeiss LSM880 microscope. In some cases, tissue sections were treated with 20 mU/mL heparinase III or 40 mU/mL chondroitinase ABC enzymes (Amsbio LLC) diluted in buffer containing 50 mM HEPES, 50 mM sodium acetate, 150 mM NaCl, and 5 mM CaCl$_2$ for 2 h at 37 °C after antigen retrieval. The subsequent steps were performed as described earlier.

## Western blotting

For Western Blot samples, mouse SMGs were immediately removed, weighed and then sonicated in RIPA Buffer (25 mM Tris, HCL pH 7.6, 150 mM NaCl, 1% NP-40, 1% sodium deoxycholate, 0.1% SDS) with protease inhibitor cocktail (Pierce No. 1860932), and phosphatase inhibitor cocktail (Millipore Sigma). Total protein concentrations were assessed using a BCA Protein Assay Kit (Pierce, Rockford, IL). NuPage™ LDS Sample Buffer and NuPage™ Sample Reducing Agent (both from ThermoFisher Scientific) were added to the homogenate, and the samples were boiled at 95 °C for 5 min. The sample (20 μg) was electrophoresed on NuPAGE™ 4–12% Bis-Tris Gels and transferred to PVDF membranes using the Invitrogen iBlot 2 Dry Blotting System (ThermoFisher Scientific). Blots were blocked with 5% BSA in 0.1% TBS-Tween (50 mM Tris-HCl pH 7.5, 150 mM NaCl, 0.1% Tween-20) for 1 h at room temperature following the transfer. The following primary antibodies were used: phospho-p44/42 MAPK (1:1000; Cell Signaling T202/Y204 #4370), p44/42 MAPK (ERK1/2) (1:1000; Cell Signaling #9102), and β-actin (1:5000, Santa Cruz #K1617). All blots were

incubated with a secondary antibody conjugated to horseradish per-oxidase (1:10,000, Cell Signaling). The blots were developed using the SuperSignal West Dura Chemiluminescent Extended Duration Sub-state (ThermoFisher Scientific). Blots were imaged on GE Amersham Imager AI680 and analyzed using NIH ImageJ software. Band inten-sities were normalized to β-actin and wildtype controls. Blots were stripped using Restore™ PLUS Western Blot Stripping Buffer (Thermo Scientific #46430) for 5 min and then blocked for 1 h with 5% BSA in 0.1% TBS-Tween.

In some cases, 50 μg lysates were diluted and treated with or without 80 mU/mL heparinase III and 100 mU/mL chondroitinase ABC diluted in buffer containing 4 mM $CaCl_2$ in PBS for 2 h at 37 °C. Lysates were boiled with Laemmli sample buffer and reducing agent and run on SDS-PAGE gels as described early. However, in this case, gels were transferred to nitrocellulose membranes using NUPAGE blotting buf-fer (ThermoFisher Scientific #NP00061) for 90 min at 95 volts. Blots were blocked with 5% milk/TBST and probed with an anti-Delta-heparan sulfate 3G10 (Asmbio #370260-1) and β-actin rabbit mono-clonal antibody-HRP conjugate (Cell Signaling #5125S).

### Lickometer testing
The licking behavior in female mice was monitored for 1 h using a lickometer (Habitest system, Coulbourn Instruments, Whitehall, PA), a computer-operated system as previously described[58]. A training ses-sion was conducted with the mice prior to the experiment, followed by a recording of the number of licks of 4 mice for 1 h. Controls and double-knockout mice were tested at the same time and then retested 5 times on separate days.

### Saliva collection
Saliva collection was performed as previously described[17]. Briefly, mice were weighed and intramuscularly anesthetized with ketamine (60 mg/kg) and xylazine (8 mg/kg). A subcutaneous injection of pilo-carpine at 0.25 mg/kg body weight was then injected to stimulate saliva secretion. Whole saliva was collected gravimetrically for 20 min using a 75-mm hematocrit tube (Drummond) into pre-weighed 1.5 mL Eppendorf tubes. The amount of saliva collected was calculated as micrograms per gram of body weight and then normalized as per-centage to the WT littermates.

### SDS-PAGE separation of salivary proteins
Saliva samples (15 μL) collected from mice were mixed with Laemmli sample buffer and reducing agent (ThermoFisher Scientific), heated at 95 °C for 5 min. These samples were then separated by gel electro-phoresis on 4–12% pre-cast bis-tris gels (ThermoFisher Scientific). The gels were stained overnight in Coomassie G-250 stain (SimplyBlue Safe stain, ThermoFisher Scientific), followed by multiple washes with deionized water to allow visualization of protein bands. Gels were imaged using GE Amersham Imager AI680.

### $[Ca^{2+}]_I$ measurements and cell volume changes
The SMGs were finely minced and digested with Liberase TL (0.5 mg/mL) twice in Eagle's minimum essential medium for 25 min each. Cells were continuously gassed with a mixture of 95% $O_2$ and 5% $CO_2$. For measurement of intracellular $Ca^{2+}$, isolated submandibular gland cells were loaded with Fura-2 (1 μM) for 45 min, and cells were excited at 340/380 nm with an emission of 510 nm. For measurement of cell volume changes, fresh dispersed SMG cells were loaded with the fluoroprobe calcein (1 μM for 10 min, Molecular Probes, Inc., Eugene, OR) and excited at 490 nm. Emitted fluorescence was measured at 510 nm. Cell volume was estimated using calcein intensity. An Olym-pus × 51 microscope (Olympus Centre Valley) with an ORCA-ER camera (Hamamatsu) attached to a Polychrome V (Till Photonics LLC) was used. Metafluor software (Molecular Devices, San Jose, CA, USA) was used for signal acquisition and data analysis. Data are presented as mean ± SE.

Origin 9 (OriginLab, Northampton, MA) was used for data analysis and display. Significant difference between individual groups was tested using ANOVA.

### FGF/FGFR pulldown assays
Streptavidin M-280 beads (ThermoFisher Scientific) (25 μL) equilibrated with 200 μL of IP-MS cell lysis buffer (ThermoFisher Scientific) were incubated with 100 μL IP-MS buffer containing 125 nM HS for 1 h on rotor at room temperature. The beads were washed three times with IP-MS cell lysis buffer. Equimolar concentrations (125 nM) of recombi-nant FGFRs and FGFs (all from R&D Systems Inc., MN) were added to the beads and incubated for 1 h on rotor at 4 °C. The supernatant was aspirated and discarded, and beads were then washed once with Sur-eQuant buffer A (ThermoFisher Scientific), followed by one wash with buffer A supplemented with 350 mM NaCl followed by one wash with buffer A supplemented with 750 mM NaCl. The beads were then washed twice with SureQuant buffer B (ThermoFisher Scientific). Washed beads were boiled with SDS-loading buffer and reducing agent and run on 4–12% bis-tris gels (ThermoFisher Scientific). The gels were stained with FASTsilver Kit (G Biosciences, #786-30) to visualize the FGFR bands. The data was obtained from at least three independent experiments. Quantification of the FGF receptor bands in the silver-stained gels was performed using the NIH ImageJ program.

### Ex vivo SMG Organ Culture
Fetal SMGs were cultured on polycarbonate membranes (13 mm, 0.1-μm pore size, Whatmann), or mesenchyme-free SMG epithelia were cul-tured in 3D Laminin-111 (Cultrex) as previously described[59,60]. Glands were photographed at 1, 24 and 48 h. For intact SMG cultures, the endbud number was counted at the beginning of the experiment (T1) and at the end of the assay at 24 h (T24) and expressed as a ratio (T24/T1) and normalized to the WT control. In some cases, fetal SMGs were dissected and immediately flash-frozen for RNA isolation. The culture medium for the isolated epithelia was supplemented with FGF10 (400 ng/mL; R&D Systems) and either unbiotinylated non-3-O-sulfated HS, 3ST1-HS, or 3ST3-HS (250 ng/mL) and cultured for 48 h at 37 °C. The morphogenic index (AU × 10$^3$) was measured as described previously[61]. Each experiment was repeated at least three times. Explants were either lysed for RNA or fixed for immunostaining.

### Myoepithelial cell culture
These cultures were performed as previously described[28]. Briefly, P2 SMGs were minced and dissociated in a 15 mL gentleMACS C tube with digestion enzymes (0.575 mg/mL collagenase type II (Gibco, USA), 1 mg/mL hyaluronidase (Sigma, USA)), 6.25 mM $CaCl_2$ (Quality Biolo-gical, USA) diluted in Hanks Buffer Salt Solution (HBSS) (Thermo Sci-entific, USA). Cell dissociation was performed in a Miltenyi gentleMACS Octo Dissociator using the manufacturer's preset 37C_h_TDK_2 program. Following dissociation, cells were washed with HBSS buffer, filtered through a 100-μm filter (Falcon, USA), and resuspended in smooth muscle cell growth media (Cell Applications, Inc., USA). The cells were strained through a 70-μm filter (MACS Mil-tenyi Biotec, USA), and 50,000 cells were plated on 24-well BioCoat™ Collagen IV multiwell plates (Corning, USA, #08–774–29) in Smooth muscle growth media (Cell Applications Inc., San Diego, CA, USA, #311–500). After 24 h, the media was removed and replaced with Smooth muscle differentiation media (Cell Applications Inc., San Diego, CA, USA, #300D-250) for the duration of the experiment. MECs were collected for qPCR after 7 days of culture.

In some cases, MECs were cultured in 8-well collagen IV-coated 15-m slide chambers (Ibidi, Fitchburg, Wisconsin, USA, #50-305-885), and after 7 days in culture, MECS were fixed with ice-cold acetone: methanol (1:1) for 10 min at −20 °C, washed with PBST. Whole-mount staining was performed with SMA (Millipore Sigma, St Louis, MO, USA. #A2547), CNN1 (Abcam, ab46794), and NGF (Alomone Labs #AN-240)

antibodies and Hoechst, and secondary antibodies as previously described[28]. Confocal stacks (3 μm stack, 0.3 μm steps) of 6–12 areas were captured on a Nikon A1Rplus MP microscope using the ×20 objective (SMA WT: $n = 10$, DKO: 12, CNN1: $n = 5$, DKO: $n = 6$, NGF WT: 5, DKO: $n = 6$). Quantification of integrated density for each channel normalized to nuclear stain was done on maximum projections using the measure analysis in FIJI.

## Statistics and reproducibility

Data were log-transformed and analyzed with an unpaired Student's $t$-test (two-tailed) or one-way ANOVA with post hoc Dunnett's multiple comparisons test when comparing more than two groups. DKO samples were compared to the WT using Prism 10 software (GraphPad, La Jolla, CA, USA). Paired two-tailed $t$-test was used for HS analysis of the male SMGs compared to the respective WT. Tables of DEGs were generated using the DESeq2 package and the non-parametric Wald test with Benjamini–Hochberg adjusted $p$ values <0.05 and Log2 fold change. The $p$-values for KEGG analysis tables are derived using right-tailed Fisher's exact $t$-test followed by a Benjamini–Hochberg adjustment.

Values are expressed as mean ± standard error of the mean (SM) from three or more biological replicates (as indicated). Images of IHC are representative, and each staining was repeated with at least three biological replicates showing similar results. Western blots and silver-stained gel images are representative, and each assay was repeated with at least three biological replicates or three independent samples.

## Reporting summary

Further information on research design is available in the Nature Portfolio Reporting Summary linked to this article.

## Data availability

The heparan sulfate mass spectrometry data for WT, DKO, WT-1, *Hs3st1* KO, *Hs3st3a1* KO and *Hs3st3b1* KO male SMGs samples generated in this study have been deposited in the Figshare database under accession code https://doi.org/10.6084/m9.figshare.26381392 [https://figshare. com/articles/dataset/MS_data_of_SMG_analysis_for_WT_Hs3st1_KO_ Hs3st3a1_KO_Hs3st3b1_KO_WT_and_DKO-_samples_from_male_SMGs_ zip/26381392?file=47954335]. The heparan sulfate mass spectrometry data for male and female SMG comparison generated in this study have been deposited in the Figshare database under accession code https:// doi.org/10.6084/m9.figshare.26381395 [https://figshare.com/articles/ dataset/MS_data_for_the_male_and_female_SMG_comparison_zip/ 26381395?file=47954338]. RNA-seq data generated in this study have been deposited in the Gene Expression Omnibus (GEO) website under accession number GSE235187. E16 scRNAseq used in this study is from previously deposited data from publicly available scRNAseq data of embryonic SMGs (GSE150327, https://sgmap.nidcr.nih.gov). Specific values used to generate graphs in this paper are provided in the Source data file. Source data are provided with this paper.

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

## Acknowledgements

We would like to thank the NIDCR Gene Transfer Core (ZIADE-000744) for generating the double-knockout mice, the NIDCR Imaging Core for help with imaging analysis (ZIC DE000750-01), the NIDCR Veterinary Resources Core for assistance with the murine procedures (ZIC DE000740-05), Combined Technical Core (ZIC DE000729), NIDCD/NIDCR Genomics and Computational Biology Core (ZIC DC000086) for preparing the RNA sequencing (RNA-seq) libraries and performing RNA-seq analysis. This work utilized the computational resources of the NIH HPC Biowulf cluster. (http://hpc.nih.gov). This project was funded in part by the Intramural Program of the NIH at NIDCR, Bethesda, MD, USA (ZIA DE000722 to M.P.H). Work in J.L. laboratory was supported by R44GM142304 to Z.W.

## Author contributions

Conceptualization, M.P.H., and V.N.P.; Methodology, M.P.H., V.N.P., Z.W., Y.X., X.L., J.L., and D.M.; Investigation, V.N.P., S.H.C., J.R.B., E.D.L., C.Z., X.L., Z.W., J.Y.P., D.M., Y.X., M.H.A., and M.P.; Resources, M.P.H., J.L., T.H.v.K, A.B.K., and I.S.A.; Data analysis, V.N.P., S.H.C., J.R.B., E.D.L., M.H.A., D.M., Z.W., X.L., and M.P.H.; Writing—Review and Editing, V.N.P.,

M.H.A., and M.P.H.; Visualization, V.N.P. and M.P.H.; Supervision, M.P.H., and V.N.P.; Funding Acquisition, M.P.H. and J.L.

## Funding

## Competing interests
J.L. is a founder and chief scientific officer for Glycan Therapeutics Corp, and Y.X. is a founder of Glycan Therapeutics. Z.W. is an employee of Glycan Therapeutics Corp and has equity in the company. The other authors declare no competing interests.
