## [Peer Review File · Nature Communications]

REVIEWER COMMENTS

Reviewer #1 (Remarks to the Author):

The manuscript by Patel et al. describes how the knockout of both Hs3st31a and Hs3st31b affects salivary gland development. The group has had a long-standing interest in the role of heparan sulfate (HS) 3-O-sulfation in this process and have published several interesting publications on the subject. Single knockouts of the Hs3st3 genes have previously been studied, and Patel et al. have also reported that HS 3-O-sulfated by Hs3st3 is more efficient in increasing myoepithelial cell differentiation than HS-3-O-sulfated by Hs3st1.

The present manuscript is somewhat difficult to read and it is not always apparent what can be learnt from the results presented compared to results obtained in previous studies. Since the double knockout has not previously been characterized it would be important to compare the phenotypes of single and double knockouts and the structure of HS in the different models. It would also be important to show if the double knockout affects expression of other HS biosynthesis enzymes (3-O-sulfotransferases as well as other enzymes). The preliminary characterization of the Hs3st1/Hs3st3b double knockout, now mentioned in the Discussion, is of course interesting but should maybe be saved to a separate publication.

Specific major and minor points

1. That Hs3st3a1 and Hs3st3b1 transcripts are enriched in MECs is stated on l. 80 and l. 161, but no reference is given.
2. The reference to the Hs3st3a1 and b1 single knockouts, Patel et al. ref. 14?, is missing on l. 100
3. On l. 73 it is stated that submandibular glands express Hs3st1/Hs3st3b 3st1, 3st3a, 3st3b, 3st6. Which 3-O-sulfotransferases are expressed in the DKO?
4. In the introduction, the difficulties of determining 3-O-sulfation are discussed and it is mentioned that these have recently been overcome (l. 75). Would have been good with a reference here, even if they are given later in the Results section (l. 124). However, in a paper by Karlsson et al. (2021) *Sci. Adv.* 7 eabl6026, these authors solve the problem by increasing the concentration of heparinase enzymes and are able to digest the more resistant structures to disaccharides. In the Karlsson paper, the substrate specificities of the 3-O-sulfotransferases are studied. They claim that the two Hs3st3 enzymes do not require the 2-O-sulfation on the Δ UA-GlcNS(6S) to add the 3-O-sulfate group. Looking at the disaccharides recovered in the present study (Fig. 1c), the levels of Δ UA-GlcNS are increased in DKO, but not marked as significant. Comments?
5. According to Fig. 1b, the DKO mice are 25% smaller in size while the SMG:s are larger. How does this compare to the single knockouts?
6. Fig. 2c: Were there no changes in the expression of HS biosynthesis enzymes?
7. If HS chains capable of binding to growth factors such as FGFs are present on cell surface proteoglycans they may enhance signaling. However, if present on ECM proteoglycans these HS may reduce growth factor signaling. Can you distinguish between cell surface and ECM location in your LACE

assays (Fig. 3a and b)?

8. The western blot in Fig. 3d is not convincing. There are no red arrows as written in the figure legend and no information on which samples that were analyzed. Did you include controls that were not treated with heparinase and loading controls?

9. The findings of increased pERK in the male acinar cells could mean that FGFR signaling is increased but could also be due to activation of any of the several other growth factor receptors signaling through the MAPK pathway (Fig. 4). Comments?

10. In suppl. Fig 5, only expression levels for Hs3st1 are shown but there is no data for the expression of Hs3st6, Hs2st1 and Hs6st1 (l. 274). In Fig. 5, all four of the are shown. What about gene expression of the other 3-O-sulfotransferases or other biosynthesis enzymes taking part in HS biosynthesis (see point 6)?

11. It is interesting that adding the HS 12mers to E13 SMGs cultured for 48 h results in different size of the epithelia depending on the presence and position of a single 3-O-sulfate group (Fig. 7c). However, the 12mers are highly sulfated and are more similar to a heparin than a HS fragment and most likely a structure that never would be encountered by the SMGc in vivo. An even more convincing result demonstrating that 3-O-sulfation of HS by Hs3st3 instead of Hs3st1 is more efficient in increasing myoepithelial cell differentiation was published by the group in 2014 (Patel et al. Dev. Cell), referred to in my summary above. In this paper, the group showed that bovine kidney HS, 3-O-sulfated by Hs3st3, is more efficient in increasing myoepithelial cell differentiation than HS 3-O-sulfated by Hs3st1. The added HS incubated with the SMGs in this experiment is probably more similar to the HS that the SMGs would be stimulated by in vivo.

12. It is unclear how the experiment in Fig. 7b was performed. Are the interactions really stable in 1 M NaCl? Also, in the SDS-PAGE the complexes should be dissolved meaning that it would be possible to both see the receptors and the FGF-ligands? Is it the receptors that are displayed in the figure and did you also quantify the FGF-ligands?

13. Finally, how come the DKO SMGs in Fig. 5a have more buds than WT SMGs, when 3-O-sulfation in Fig. 7C is stimulated by 3-O-sulfation?

Reviewer #2 (Remarks to the Author):

This primary research study by Patel and colleagues investigates heparan sulfate (HS) regulation of FGFR in the developing and adult salivary gland, using complex knockout mouse models and primary explant culture assays, ultimately finding that highly 3-O-sulfated HS domains regulate FGFR signalling to impact myoepithelial cells, and that in the absence of Hs3st3a1 and Hs3stb1 epithelial integrity is abnormal. The study is well designed, uses elegant approaches and provides robust and novel data. Overall, the study is a clear addition to the field. I have the following comments that should be addressed before publication.

Introduction

Introduction, paragraph 1: I believe it is also worth mentioning that FGF signalling is not only a target for regenerative medicine, but also for human conditions associated with abnormal FGF signalling, where salivary gland developmental defects are observed (e.g. lacrimo-auriculo-dento-digital syndrome (LADD); PMID: 16630169). This can only strengthen the importance of this area of study.

Introduction: could the authors please specify why the fact that the SMG only expresses 4 of the 7 Hs3st sulfotransferases makes it a good model to study (rather than a situation where all 7 are expressed)? (line 72)

Introduction: could the authors please make the citation for the statement “A major issue, recently overcome, was the biochemical analysis of 3-OHS, and how to directly measure whether a genetic deletion of specific 3-O-sulfotransferase reduces 3-O-HS in tissue.” (page 3, line 75) clear (if published). i.e. has the technique by which this has recently been overcome been published in a manner that the authors could cite?

Introduction: Regarding the statement “MECs are progenitors during development, homeostasis, and regeneration” (page 4, line 79) – could the authors elaborate on the progenitor potential of MECs (i.e. progenitor cells for what lineage?) and the fact that this potential is likely different in homeostasis compared to injury. A study from 2018 (PMID 30305288) showed, via lineage tracing under the control of an Acta2 promoter, that MECs give rise to themselves via self-duplication during homeostasis, but not to any other cell lineage. However, following this, a 2020 study (PMID: 32994165) found, using a similar lineage tracing method, that following injury, MECs contribute to acinar and ductal cells. Thus, MECs have some level of progenitor plasticity depending on the need for cell replacement.

Results

Results (page 5, line 100): It would be worth stating in the text what the background for the recently generated Hs3st3a1 and Hs3stb1 KO mice is (e.g. “We previously generated Hs3st3a1 and Hs3st3b1 knockout (KO) mice on a XXX background, and have also obtained Hs3st1 KO mice on a C57Bl6 background.”

Minor: technically, C57Bl6 should be C57BL/6

Figure 1: How many independent litters were used for quantification in Figure 1A?

Figure 1: Is it possible to present the body weight and SMG weight data as individual data points in addition to the mean (bar) and error bars and present the n number analysed here in the figure legend. In addition, what does the error bar show (SD or SEM)? Please also include this information in the figure legend, alongside the statistical test used.

Figure 2: thank you for including both male and female SMG in your analysis and showing the sexual dimorphism in epithelial architecture.

Figure 2: could the differences in epithelia between the WT and DKO be quantified please? e.g. number of granular ducts per field of view, number of acini per field of view. This should be relatively straightforward to do based on the cell morphology, using nuclei as a measure of an individual cell. This will strengthen the conclusion that DKO SMG morphology is changed.

Figure 2: if possible, please present the data as individual data points, in addition to the mean (bar) and error bars.

Figure 4: is there any way to quantify the staining data presented in figure 4b? This would strengthen conclusions such as “duct staining appeared reduced” (page 9, line 226)

Figure 4b: does the phosphatase treatment abolish all staining in the WT but just pErk in the DKO? Is that what would be expected (i.e. completely blank image)?

Figure 5a: please include scale bars on the brightfield images.

Figure 7: I suspect that “Immunostaining for MEC markers SMA and CNN1 (Fig. 7d)” should be 7e rather than 7d?

Materials and Methods

Page 20, line 541: Could the authors please also provide the dilution of antibodies used for immunofluorescent staining?

Ref.: Nature Communications manuscript NCOMMS-24-02038

Title: Specific 3-*O*-sulfated heparan sulfate domains regulate salivary gland basement membrane metabolism and epithelial differentiation

Response to Reviewers:

We would like to thank the reviewers for their helpful suggestions and comments. We have addressed all of them below, highlighting the changes that have been made to the text in the manuscript.

Reviewer #1 (Remarks to the Author):

The manuscript by Patel et al. describes how the knockout of both Hs3st31a and Hs3st31b affects salivary gland development. The group has had a long-standing interest in the role of heparan sulfate (HS) 3-*O*-sulfation in this process and have published several interesting publications on the subject. Single knockouts of the Hs3st3 genes have previously been studied, and Patel et al. have also reported that HS 3-*O*-sulfated by Hs3st3 is more efficient in increasing myoepithelial cell differentiation than HS-3-*O*-sulfated by Hs3st1.

The present manuscript is somewhat difficult to read and it is not always apparent what can be learnt from the results presented compared to results obtained in previous studies. Since the double knockout has not previously been characterized it would be important to compare the phenotypes of single and double knockouts and the structure of HS in the different models. It would also be important to show if the double knockout affects expression of other HS biosynthesis enzymes (3-*O*-sulfotransferases as well as other enzymes). The preliminary characterization of the Hs3st1/Hs3st3b double knockout, now mentioned in the Discussion, is of course interesting but should maybe be saved to a separate publication.

Reply: We appreciate the reviewer's comments and agree that these suggestions will improve the paper. We have clarified the text to highlight differences between single and double KO mice, and some of these changes are detailed here.

We have also included new data comparing the HS structures by heparan sulfate analysis of the disaccharides and tetrasaccharides from the *Hs3st1*, *Hs3st3a1*, and *Hs3st3b1* single knockout mice in **Supplementary Figure 1d and 1e**. The text from page 5, line 132 reads:

“We analyzed both non-3-*O*-sulfated disaccharides and 3-*O*-sulfated tetrasaccharides in the SMGs from DKO and the single KOs of *Hs3st1*, *Hs3st3a1* and *Hs3st3b1*. (Fig. 1c, 1d and supplementary Fig. 1d and 1e). Disaccharide analysis showed a significant increase in Δ UA2S-GlcNS and an increasing trend for Δ UA-GlcNS while other disaccharides showed no differences in the DKO SMGs compared to the WT (Fig. 1c). Disaccharide analysis in the single knockouts did not show any differences compared to the WT control (Supplementary Fig. 1d).

Interestingly, tetrasaccharide analysis resolved 5 different 3-*O*-sulfated tetrasaccharides in WT HS. One tetrasaccharide, Δ UA-GlcNS6S-IdoA2S-GlcNS3S6S, was not detected in the DKO, *Hs3st3a1* KO and *Hs3st3b1* KO (Fig. 1d, supplementary Fig. 1e). This most highly sulfated

tetrasaccharide is predicted to be the product of HS3ST3 enzymes. In addition, Δ UA-GlcNS-IdoA2S-GlcNS3S, which is also predicted to be a product of HS3ST3 enzymes, was not significantly decreased in the DKO, *Hs3st3a1*, or *Hs3st3b1* KO SMGs. The tetrasaccharide, Δ UA-GlcNS6S-GlcA-GlcNS3S6S, containing 3-O-sulfation correlating to the antithrombin binding site proposed to be generated by HS3ST1 significantly decreased in the DKO HS (Fig. 1d), but not in the *Hs3st3a1*, *Hs3st3b1* and *Hs3st1* KO SMGs compared to WT. Interestingly, Δ UA-GlcNAc6S-GlcA-GlcNS3S6S was not detected in the SMGs of *Hs3st1* KO SMGs (Supplementary Fig. 1e) whereas no differences were detected in Hs3st3 KO mice.”

Figure legend for supplementary Figure 1d and 1e reads:

“**d-e** Mass spec analysis of HS disaccharides and tetrasaccharides in adult SMGs. Analysis of HS disaccharide **d**, and tetrasaccharide **e** composition between WT and *Hs3st1*, *Hs3st3a1* (3A1) and *Hs3st3b1* (3B1) single KO SMGs. All data shown is averaged. N=3 consisting of 3-4 SMGs pooled together for each group. Δ UA2S-GlcNAc6S not detected in SMGs. Δ UA-GlcNS6S-IdoA2S-GlcNS3S6S not detected in *Hs3st3a1* (3A1) and *Hs3st3b1* (3B1) SMGs. Δ UA-GlcNAc6S-GlcA-GlcNS3S6S not detected in *Hs3st1* KO.”

Gene expression analysis of the heparan sulfate biosynthetic enzymes in the double knockout mice has been performed for the adult SMGs (**Figure 1e**), embryonic gland E13 (**Supplementary Figure 5**) and E13 SMGs cultured for 24h (**Figure 5b**), highlighting that only slight, but not significant, transcriptional changes of enzyme expression occur.

Please see point 3 and point 10 below for specific changes to the text.

We thank the reviewer for the comment regarding the characterization of the *Hs3st1*/*Hs3st3b* double knockout. We agree that this information is interesting and important to researchers in the heparan sulfate field who are trying to generate novel knockouts. We have therefore left the information in the discussion section.

Specific major and minor points

1. That *Hs3st3a1* and *Hs3st3b1* transcripts are enriched in MECs is stated on l. 80 and l. 161, but no reference is given.

Reply: The reference has now been included in the text.

2. The reference to the *Hs3st3a1* and *b1* single knockouts, Patel et al. ref. 14?, is missing on l.100

Reply: This reference is now included.

3. On l.73 it is stated that submandibular glands express *Hs3st1*/*Hs3st3b* *3st1*, *3st3a*, *3st3b*, *3st6*. Which 3-O-sulfotransferases are expressed in the DKO?

Reply: The DKO SMGs expresses *Hs3st1*, and *Hs3st6*. The expression of *Hs3st2*, *Hs3st4* and *Hs3st5* is not detected in both the adult WT and DKO SMGs. The gene expression analysis of other heparan sulfate biosynthetic enzymes in the double knockout mice has now been included in **Figure 1e**.

The text in the results section page 6, line 150 now reads:

“Quantitative PCR analysis in the adult DKO SMGs showed, as expected, no expression of *Hs3st3a1* and *Hs3st3b1*, while other HS enzymes in DKO SMGs were comparable to the WT (Fig.

1e). Both WT and DKO adult SMG had no detection of *Hs3st2*, *Hs3st4*, *Hs3st5*, *Hs6st3*, and *Ndst4*.”

The figure legend for **Figure 1e** on page 28, line 801 reads “e Gene expression changes in HS biosynthetic enzymes in both male and female DKO SMGs normalized to WT and *Rps29*. Error bars are SM (n>3). Unpaired t-test compared to WT, ***p < 0.001.”

4. In the introduction, the difficulties of determining 3-O-sulfation are discussed and it is mentioned that these have recently been overcome (l. 75). Would have been good with a reference here, even if they are given later in the Results section (l. 124). However, in a paper by Karlsson et al. (2021) *Sci. Adv.* 7 eabl6026, these authors solve the problem by increasing the concentration of heparinase enzymes and are able to digest the more resistant structures to disaccharides. In the Karlsson paper, the substrate specificities of the 3-O-sulfotransferases are studied. They claim that the two Hs3st3 enzymes do not require the 2-O-sulfation on the DUA-GlcNS(6S) to add the 3-O-sulfate group. Looking at the disaccharides recovered in the present study (Fig. 1c), the levels of DUA-GlcNS are increased in DKO, but not marked as significant. Comments?

Reply: We thank the reviewer for pointing this out. We have now added the references for the discussion of the difficulties of determining 3-O-sulfation in the introduction of the manuscript on page 4 line 73 and now reads:

“A major issue, recently overcome, was the biochemical analysis of 3-O-HS since 3-O-sulfation can cause resistance to enzyme digestion, and how to directly measure whether a genetic deletion of specific 3-O-sulfotransferase reduces 3-O-HS in tissue¹⁸⁻²⁰. Increasing the concentration of heparinase enzymes has also recently enabled digestion of the more resistant structures to 3-O-sulfated disaccharides²¹. In addition, recent chemoenzymatic synthesis of defined 3-O-HS structures allow us to investigate their function. These advances allow more specific probing into the study of the 3-O-sulfated structure and function relationship of heparan sulfate.”

In addition, we have removed the statement “A 2S on a UA is required for HS3ST3 enzymes to add a 3-O-sulfate, so this suggests with loss of the 3-O-sulfotransferase there is an increase in 2-O-sulfated disaccharides, which would be substrates for the enzymes.” to be consistent with the paper by Karlsson et al., (2021).

Statistical analysis of the UA-GlcNS shows an increasing trend, however, is not significantly different in the DKO and the WT. The text on page 6, line 135 is now amended as indicated in the first comment above.

5. According to Fig. 1b, the DKO mice are 25% smaller in size while the SMGs are larger. How does this compare to the single knockouts?

Reply: The salivary glands from adult *Hs3st3a1* and *Hs3st3b1* single knockouts are similar in size to wildtype. This is now incorporated in the text on page 5 line 122 and reads:

“In contrast to the single KO strains¹⁷, DKO mice were ~ 25% smaller in size than their WT littermates (Fig. 1b). Further, gross anatomic analyses of adult SMGs showed relatively larger glands in DKO compared to WT when normalized to the body weights (Fig. 1b). Taken together, this suggested that DKO mice have less functional redundancy than single KO strains.”

6. Fig. 2c: Were there no changes in the expression of HS biosynthesis enzymes?

Reply: The gene expression analysis of the heparan sulfate modifying biosynthetic enzymes in the double knockout mice has now been included in **Figure 1e**. Please see point 3 for detailed edits of the text.

7. If HS chains capable of binding to growth factors such as FGFs are present on cell surface proteoglycans they may enhance signaling. However, if present on ECM proteoglycans these HS may reduce growth factor signaling. Can you distinguish between cell surface and ECM location in your LACE assays (Fig. 3a and b)?

Reply: The reviewer makes an interesting point here. With the limitations of the assay, we are not able to distinguish between cell surface and ECM location in the LACE assay. However, with colocalization of the FGF10/FGFR2b complex with the basement membrane protein, COLIV, it can be speculated that the complex is binding to the basement membrane. To comment on the cell surface HSPGs, the specific 3-*O*-sulfated HSPGs would have to be identified, furthermore cell sheddases may release cell surface HSPGs into the extracellular space complicating this type of analyses. In response to the reviewer's comment, the statement has been rephrased in the text on page 8, line 216 to read:

“Surprisingly, the DKO had increased binding of the FGF10/FGFR2b complex surrounding acinar structures and ducts, overlapping with increased type-IV collagen (ColIV) staining in the BM compared to WT (Fig. 3a).”

The associated figure legend for Figure 3 on page 29, line 823 reads:

“Fig. 3. DKO SMGs have increased binding of FGF10/FGFR2b-Fc complex (LACE) overlapping with the basement membrane and increased collagen IV, and HSPGs (3G10 staining) associated with acinar cells.

a Binding of FGF10/FGFR2b-Fc complex overlapping with COLIV, a basement membrane protein, is increased in DKO SMGs.”

Also, text on Line 230 on page 9 is clarified and now reads:

“..... 3G10 staining also increased surrounding the mucin10+ acinar structures in the DKO (Fig. 3b and 3c).”

8. The western blot in Fig. 3d is not convincing. There are no red arrows as written in the figure legend and no information on which samples that were analyzed. Did you include controls that were not treated with heparinase and loading controls?

Reply: We apologize for the errors made in the earlier version of Fig. 3d. The Figure is now updated to include lysates not treated with the heparinase III enzyme and loading control.

Figure legend associated with Figure 3d on page 29, line 834 reads:

“d Western blot of 3G10 antibody, which detects the HS stub, shows increase in some HSPG cores (indicated by red arrows) in DKO SG lysate treated with or without heparinase III. β -actin is also shown.”

9. The findings of increased pERK in the male acinar cells could mean that FGFR signaling is increased but could also be due to activation of any of the several other growth factor receptors signaling through the MAPK pathway (Fig. 4). Comments?

Reply: We agree with the reviewer that this is possible and have amended the text on page 9, line 249 to read:

“In addition, increased MAPK signaling could also be activated via growth factor receptors other than FGFRs signaling.”

10. In suppl. Fig 5, only expression levels for Hs3st1 are shown but there is no data for the expression of Hs3st6, Hs2st1 and Hs6st1 (l. 274). In Fig. 5, all four of the are shown. What about gene expression of the other 3-O-sulfotransferases or other biosynthesis enzymes taking part in HS biosynthesis (see point 6)?

Reply: We thank the reviewer for pointing out the inconsistency in genes screened for qPCR analysis of the E13 SMGs cultured for 24h (Fig. 5b) and the freshly dissected E13 SMGs (Supplementary Fig. 5.) The qPCR analysis for other HS modifying biosynthetic enzymes is now included in **Fig. 5b** and **Supplementary Fig. 5**.

The text in the results on page 11, line 296 reads:

“In the DKO embryonic day (E) 13 SMG, when branching morphogenesis begins, there were no compensatory transcriptional changes in other 3-O-sulfotransferase enzymes (*Hs3st1*, *Hs3st2*, *Hs3st4*, *Hs3st5* and *Hs3st6*), although expression of *Hs3st2*, *Hs3st4* and *Hs3st5* is barely detectable in the SMGs (Supplementary Fig. 5). As expected, *Hs3st3a1* and *Hs3st3b1* were not detected while other HS biosynthetic genes did not change in DKO SMGs compared to WT (Supplementary Fig. 5).”

Page 11, Line 310 now reads “Gene expression analysis of the 3-O-sulfotransferase transcripts showed *Hs3st3a1* and *Hs3st3b1* were not detected and there was a decrease in *Hs3st2*. No other changes in gene expression were detected for any other HS biosynthetic genes analyzed (Fig 5b).”

11. It is interesting that adding the HS 12mers to E13 SMGs cultured for 48 h results in different size of the epithelia depending on the presence and position of a single 3-O-sulfate group (Fig. 7c). However, the 12mers are highly sulfated and are more similar to a heparin than a HS fragment and most likely a structure that never would be encountered by the SMGc in vivo. An even more convincing result demonstrating that 3-O-sulfation of HS by Hs3st3 instead of Hs3st1 is more efficient in increasing myoepithelial cell differentiation was published by the group in 2014 (Patel et al. Dev. Cell), referred to in my summary above. In this paper, the group showed that bovine kidney HS, 3-O-sulfated by Hs3st3, is more efficient in increasing myoepithelial cell differentiation than HS 3-O-sulfated by Hs3st1. The added HS incubated with the SMGs in this experiment is probably more similar to the HS that the SMGs would be stimulated by in vivo.

Reply: We thank the reviewer for the comment. However, in the paper Patel et al., 2014, we showed that Hs3st3-modified HS expanded the KIT+FGFR2+ progenitor during early SMG development, prior to myoepithelial differentiation. Therefore, the markers for myoepithelial cells were not screened.

We agree with the reviewer that the 12mers used in the current study are more like a heparin than a HS fragment. However, using the 12mer HS helped us dissect the effects of Hs3st1 and Hs3st3-modified HS structures since these reagents are chemoenzymatically synthesized in vitro and have a defined length, known sulfation levels and chemical structures. The previously used

bovine HS is a mixture of polysaccharides with different chain sizes and sulfation patterns.

12. It is unclear how the experiment in Fig. 7b was performed. Are the interactions really stable in 1 M NaCl? Also, in the SDS-PAGE the complexes should be dissolved meaning that it would be possible to both see the receptors and the FGF-ligands? Is it the receptors that are displayed in the figure and did you also quantify the FGF-ligands?

Reply: We apologize for lack of detail in the protocol for the pulldown assays. The description of the protocol is improved and under the methods sections for FGF/FGFR Pulldown assays on page 25, line 715 reads:

“Streptavidin M-280 beads (ThermoFisher Scientific) (25 uL) equilibrated with 200 uL of IP-MS cell lysis buffer (ThermoFisher Scientific) were incubated with 100 uL IP-MS buffer containing 125 nM HS for one hour on rotor at room temperature. The beads were washed three times with IP-MS cell lysis buffer. Equimolar concentrations (125 nM) of recombinant FGFRs and FGFs (all from R&D Systems Inc., MN) were added to the beads and incubated for one hour on rotor at 4°C. The supernatant was aspirated and discarded, and beads were then washed once with SureQuant buffer A (ThermoFisher Scientific), followed by one wash with buffer A supplemented with 350mM NaCl followed by one wash with buffer A supplemented with 750mM NaCl. The beads were then washed twice with SureQuant buffer B (ThermoFisher Scientific). Beads were then boiled with SDS-loading buffer and reducing agent and run on 4-12% bis-tris gels (ThermoFisher Scientific). The gels were stained with FASTsilver Kit (G Biosciences, #786-30) to visualize the FGFR bands. The data was obtained from at least three independent experiments. Quantification of the FGF receptor bands in the silver-stained gels was performed using the NIH ImageJ program.”

Salt washes with increasing NaCl concentration were performed for the complex binding of FGFR1b and FGF1 to determine the most ideal salt wash to eliminate any non-specific binding of the receptors to the Streptavidin beads without HS. This experiment had shown that the receptor and ligand bound non-specifically to the beads at washes supplemented with less than 350mM NaCl (500mM salt wash). The buffer A supplemented with 750mM NaCl (1M salt wash) was ideal to detect the FGFR1b differences between the three HS, although at this concentration, the detection of FGF ligands was much weaker than the receptors.

We agree that in the SDS-PAGE the complexes of receptors and ligands will be dissolved. The blots in Fig 7b show the receptors. The detection and quantification of the FGF ligands was not feasible since the ligand bands were weakly detected by this method.

13. Finally, how come the DKO SMGs in Fig. 5a have more buds than WT SMGs, when 3-O-sulfation in Fig. 7C is stimulated by 3-O-sulfation?

Reply: These results reflect the different types of assay and model systems used for the experiments and highlights the complexity of identifying functions of specific HS sulfation at discrete stages of development. We explain the increased number of buds in the intact DKO SMG in Fig.5a, as being related to increased diffusion of growth factors such as FGF10, when the HS in the ECM has less 3-O-sulfation. Thus, losing both the *Hs3st3a1* and *Hs3st3b1* enzymes and thus less 3-O-sulfation, no longer restricts growth factor signaling at discrete sites

near the epithelial tips and increases signaling throughout the endbud resulting in an increasing, although not significant, trend of more buds in the DKO SMG (Fig. 5a).

In contrast, Figure 7c is an assay with no endogenous mesenchyme, where isolated SMG epithelia are cultured in laminin-111 extracellular matrix (ECM), which contains perlecan heparan sulfate (Patel et al., 2007, *Development*), and only recombinant FGF10, and due to the high binding of FGF10 to HS, FGF10 binds to the ECM HS mediating short range diffusion, limiting the action of FGF10 to the tips of the epithelial buds and thus displays a ductal morphology (Makarenkova et al., 2009, *Science Signaling*). In the presence of exogenous 3-O-sulfated HS, FGF10 can be competed from the ECM and is able to diffuse more freely and results in increased epithelial branching. At the concentrations of the 12mer HS used in this assay, addition of HS3ST3-HS increased epithelial branching suggesting that application of FGF10 and HS3ST3-HS mediates broader diffusion range than non-3-O-sulfated HS (Fig. 7c).

Reviewer #2 (Remarks to the Author):

This primary research study by Patel and colleagues investigates heparan sulfate (HS) regulation of FGFR in the developing and adult salivary gland, using complex knockout mouse models and primary explant culture assays, ultimately finding that highly 3-O-sulfated HS domains regulate FGFR signalling to impact myoepithelial cells, and that in the absence of Hs3st3a1 and Hs3stb1 epithelial integrity is abnormal.

The study is well designed, uses elegant approaches and provides robust and novel data. Overall, the study is a clear addition to the field. I have the following comments that should be addressed before publication.

Introduction

Introduction, paragraph 1: I believe it is also worth mentioning that FGF signalling is not only a target for regenerative medicine, but also for human conditions associated with abnormal FGF signalling, where salivary gland developmental defects are observed (e.g. lacrimo-auriculo-dento-digital syndrome (LADD); PMID: 16630169). This can only strengthen the importance of this area of study.

Reply: We thank the reviewer for this point and the text is revised to include this information and on page 3, line 41 now reads:

“FGFR signaling is required for development, function, and regeneration in a wide range of organs including salivary glands (SG)^{1,2}. The essential role of FGFR signaling in SGs is highlighted by mutations in FGF10 or FGFR2b which lead to two rare genetic disorders; aplasia of lacrimal and salivary glands (ALSG: MIM #180920) and lacrimo-auriculo-dento-digital syndrome (LADD: MIM #149730)³. In addition, SG dysfunction due to irradiation damage following head and neck cancer therapy or after autoimmune destruction in Sjögren’s disease remain major clinical challenges and FGFR signaling is an attractive target for regenerative strategies^{4,5}.”

Introduction: could the authors please specify why the fact that the SMG only expresses 4 of the 7 Hs3st sulfotransferases makes it a good model to study (rather than a situation where all 7 are expressed)? (line 72)

Reply: The functional redundancy among the seven enzyme isoforms could complicate interpretation of results from a knockout model tissue where all isoforms are expressed. We have revised the text to state this more clearly, and the text from page 3, line 68 now reads as follows: “Seven enzyme isoforms generate the 3-*O*-sulfated domain and functional redundancy exists among them. The submandibular gland (SMG) epithelia only express four of the seven Hs3st sulfotransferases (*Hs3st1*, *Hs3st3a1*, *Hs3st3b1*, *Hs3st6*) in a spatial and temporal manner^{15,17}. The expression of fewer enzyme isoforms reduces potential functional redundancy in the tissue making the SMG a useful model to investigate 3-*O*-sulfation.”

Introduction: could the authors please make the citation for the statement “A major issue, recently overcome, was the biochemical analysis of 3-OHS, and how to directly measure whether a genetic deletion of specific 3-O-sulfotransferase reduces 3-O-HS in tissue.” (page 3, line 75) clear (if published). i.e. has the technique by which this has recently been overcome been published in a manner that the authors could cite?

Reply: We thank the reviewer for pointing out the missing citations. The biochemical analysis has recently been published and we have now included the following references to this sentence on page 4, line 73: “A major issue, recently overcome, was the biochemical analysis of 3-*O*-HS since 3-*O*-sulfation can cause resistance to enzyme digestion, and how to directly measure whether a genetic deletion of specific 3-*O*-sulfotransferase reduces 3-*O*-HS in tissue¹⁸⁻²⁰. Increasing the concentration of heparinase enzymes has recently enabled digestion of the more resistant structures to 3-*O*-sulfated disaccharides²¹. In addition, recent chemoenzymatic synthesis of defined 3-*O*-HS structures allow us to investigate their function.”

18 Wang, Z. et al. Correction to Analysis of 3-*O*-Sulfated Heparan Sulfate Using Isotopically Labeled Oligosaccharide Calibrants. *Anal Chem* 94, 4134, doi:10.1021/acs.analchem.2c00670 (2022).

19 Wang, Z. et al. Analysis of 3-*O*-Sulfated Heparan Sulfate Using Isotopically Labeled Oligosaccharide Calibrants. *Anal Chem* 94, 2950-2957, doi:10.1021/acs.analchem.1c04965 (2022).

20 Wang, Z. et al. Increased 3-*O*-sulfated heparan sulfate in Alzheimer's disease brain is associated with genetic risk gene HS3ST1. *Sci Adv* 9, eadf6232, doi:10.1126/sciadv.adf6232 (2023).

21 Karlsson, R. *et al.* Dissecting structure-function of 3-*O*-sulfated heparin and engineered heparan sulfates. *Sci Adv* 7, eabl6026, doi:10.1126/sciadv.abl6026 (2021).

Introduction: Regarding the statement “MECs are progenitors during development, homeostasis, and regeneration” (page 4, line 79) – could the authors elaborate on the progenitor potential of MECs (i.e. progenitor cells for what lineage?) and the fact that this potential is likely different in homeostasis compared to injury. A study from 2018 (PMID 30305288) showed, via lineage tracing under the control of an *Acta2* promoter, that MECs give rise to themselves via self-duplication during homeostasis, but not to any other cell lineage. However, following this, a 2020 study (PMID: 32994165) found, using a similar lineage tracing method, that following injury, MECs contribute to acinar and ductal cells. Thus, MECs have some level of progenitor plasticity depending on the need for cell replacement.

Reply: We thank the reviewer for this point and agree that the text should be elaborated further and have edited the text on page 4, line 81 as follows: “In SG, *Hs3st3a1* and *Hs3st3b1* are enriched in MECs¹⁷, which wrap around secretory acinar cells and can act as progenitors during development, homeostasis, and regeneration²²⁻²⁵. MEC produce BM proteins such as laminins, fibronectins, HSPGs and collagen IV, and are known to be regulators of tissue polarity^{26,27}. Preclinical models show that MEC have bi-potent progenitor potential following severe damage while during homeostasis, they are uni-potent and self-maintained^{22,28}. We recently showed that nerve growth factor (NGF) drives MEC differentiation during development and upregulated neurotrophin signaling in human MEC after irradiation for cancer is associated with stress-induced plasticity and lack of regeneration²⁹. MECs arise from SG endbud progenitors, which transition from a keratin14 (Krt14)+ multipotent state via two bipotent states, and to a unipotent state during homeostasis³⁰. In postnatal glands, MECs act as a signaling hub, expressing FGF7, which activates an acinar transcriptional program²⁹ involving both FGFR2¹ and Kras³⁰ signaling to drive acinar differentiation.”

Results

Results (page 5, line 100): It would be worth stating in the text what the background for the recently generated *Hs3st3a1* and *Hs3stb1* KO mice is (e.g. “We previously generated *Hs3st3a1* and *Hs3st3b1* knockout (KO) mice on a XXX background, and have also obtained *Hs3st1* KO mice on a C57Bl6 background.”

Minor: technically, C57Bl6 should be C57BL/6

Reply: We thank the reviewer for pointing this out. The text on page 5, line 106 now reads “We previously generated *Hs3st3a1* and *Hs3st3b1* knockout (KO) mice¹⁷ and have also obtained *Hs3st1* KO mice, all on a C57BL/6 background^{31 32}.”

Figure 1: How many independent litters were used for quantification in Figure 1A?

Reply: 25 independent litters derived from heterozygous crosses were quantified to determine the Mendelian ratio of the DKO mice. The legend for Fig. 1a on page 28, line 792 now reads “Data shown are the number of mice of a given genotype for each sex derived from 25 independent litters obtained from heterozygous crosses.”

Figure 1: Is it possible to present the body weight and SMG weight data as individual data points in addition to the mean (bar) and error bars and present the n number analysed here in the figure legend. In addition, what does the error bar show (SD or SEM)? Please also include this information in the figure legend, alongside the statistical test used.

Reply: We thank the reviewer for pointing out this missing information in the figure legend. Data is now presented as individual data points with error bars showing SEM. We have included the information in the figure legend for Fig. 1 and now reads on page 28, line 794 as follows: “b Body weight and SMG weight of adult male and female mice normalized to body weight and WT. Error bars are SM (n>35). Unpaired t-test. *** $P < 0.001$, compared to WT.”

Figure 2: thank you for including both male and female SMG in your analysis and showing the sexual dimorphism in epithelial architecture.

Reply: We thank the reviewer for the comment.

Figure 2: could the differences in epithelia between the WT and DKO be quantified please? e.g. number of granular ducts per field of view, number of acini per field of view. This should be relatively straightforward to do based on the cell morphology, using nuclei as a measure of an individual cell. This will strengthen the conclusion that DKO SMG morphology is changed.

Reply: We thank the reviewer for making this point. We have included quantification of the epithelia in the DKO and WT for both male and female glands. The results show an increase in number of acinar cells per area and a concomitant reduction in ducts per area in both sexes. The results are now included in **Figure 2a**.

The text on page 6, line 158 is revised as follows to include this result:

“The murine SMG is sexually dimorphic, with males having larger granular ducts with prominent pink granular cytoplasm compared to those of the female (Fig. 2a, arrows). When compared to WT, the gross morphology of DKO SMG in both sexes showed reduction in granular ducts and an increase in acinar cells per area (Fig. 2a), suggesting the epithelial progenitor niche may be affected.”

The figure legend is revised and now reads as follows from line 806 on page 28: “**a** Representative images of Hematoxylin Eosin (H&E) staining of adult male and female WT and DKO SMGs. Arrows point to granular ducts. Scale bar, 50 μ m. Quantification of ducts and acinar cells per area normalized to the WT SMGs. Graphs show Mean \pm SM. N=3-4 glands with 13-21 areas quantified (Male WT: n=21, male DKO: n=20, female WT: n=19, female DKO: n=12). Unpaired t-test. *** $P < 0.001$, compared to WT.”

Figure 2: if possible, please present the data as individual data points, in addition to the mean (bar) and error bars.

Reply: The data is now presented as individual data points with interleaved bars showing mean with error bars (SEM).

Figure 4: is there any way to quantify the staining data presented in figure 4b? This would strengthen conclusions such as “duct staining appeared reduced” (page 9, line 226)

Reply: Quantification of the pERK staining in both the acinar and ducts has been performed. Our data indicates that pERK staining in the acinar is increased but there is no difference in the staining within the ducts. We thank the reviewer for making this suggestion for quantification, which is now added to supplementary Figure 4b.

The text has been amended and on page 9, line 244 now reads “The localization of pERK in male DKO SMG was analyzed using immunostaining and showed a significant increase in pERK staining in the DKO acinar cells (Fig. 4b and supplementary Fig. 4b).”

Figure legend for supplementary Figure 4b reads “**b** Quantification of pERK staining in acini and ducts normalized to nuclei shown as fold change normalized to WT. Graph shows Mean \pm SM. Unpaired T-test compared to WT, ** $p < 0.01$.”

Figure 4b: does the phosphatase treatment abolish all staining in the WT but just pErk in the DKO? Is that what would be expected (i.e. completely blank image)?

Reply: The phosphatase treatment eliminated the specific pERK staining, but weak background

staining is still present. We thank the reviewer for this point, and we have corrected the organization of the images in Figure 4b to show this control.

Figure 5a: please include scale bars on the brightfield images.

Reply: A scale bar has been added to the brightfield images in Figure 5a.

The associated Figure legend on page 30, line 866 for Figure 5a reads “a Light micrograph of SMGs isolated from E13 embryos and cultured ex vivo for 24 hours. Graph is quantification of the number of endbuds (expressed as a ratio of the number at 24 h/the number at 1 h) of WT (n=12) and DKO (n=10) SMGs. Error bars: SM. Scale bar, 100µm.”

Figure 7: I suspect that “Immunostaining for MEC markers SMA and CNN1 (Fig. 7d)” should be 7e rather than 7d?

Reply: This error has been corrected in the result section on page 14, line 385.

Materials and Methods

Page 20, line 541: Could the authors please also provide the dilution of antibodies used for immunofluorescent staining?

Reply: This is a very good suggestion, and a **Supplementary Table 2** is now included with this submission and this table also outlines the dilutions used for the immunofluorescent staining.

REVIEWER COMMENTS

Reviewer #1 (Remarks to the Author):

The revision has greatly improved the manuscript I have two remaining issues that I would like the authors to respond to/comment on:

1. The SMG sex difference:

HS structure presented in Fig. 1 Since male and female SMGs differ it would be important to see if this difference is also reflected on heparan sulfate structure. Were the structural analyses shown in Fig.1 performed on male or female SMGs? I also wonder if the WT control used in the qPCR-assays shown in Fig 1e was male for the male samples and female for the female ones or if the same WT control was used for both. The sex difference could possibly also affect the results from the LACE assays shown in Fig. 3 and salivary secretion in Fig. 4.

2. The Western blot shown in Fig. 3:

There is a lot of background staining with the 3G10 antibody, quite similar in the four lanes. In contrast, according to the b-actin staining, there is much less sample in the heparinase-treated samples. Most importantly, the sizes of the “new” bands obtained after heparinase digestion of the DKO sample do not match any known size of HSPG core proteins. Maybe the band eluting between the 97 and 191 markers could be a syndecan with remaining chondroitin sulfate chains. A combined heparinase/chondroitinase digestion could solve this problem.

Reviewer #2 (Remarks to the Author):

The authors have addressed all my comments, thank you.

Ref.: Nature Communications manuscript NCOMMS-24-02038A

Title: Specific 3-*O*-sulfated heparan sulfate domains regulate salivary gland basement membrane metabolism and epithelial differentiation

Response to Reviewers:

We would like to thank reviewer#1 for the helpful suggestions and comments to further clarify the manuscript. We have addressed the concerns below, highlighting the changes that have been made to the text in the manuscript.

Reviewer #1 (Remarks to the Author):

The revision has greatly improved the manuscript I have two remaining issues that I would like the authors to respond to/comment on:

1. The SMG sex difference:

HS structure presented in Fig. 1 Since male and female SMGs differ it would be important to see if this difference is also reflected on heparan sulfate structure. Were the structural analyses shown in Fig.1 performed on male or female SMGs? I also wonder if the WT control used in the qPCR-assays shown in Fig 1e was male for the male samples and female for the female ones or if the same WT control was used for both. The sex difference could possibly also affect the results from the LACE assays shown in Fig. 3 and salivary secretion in Fig. 4.

Reply: The HS analysis shown in Fig. 1 and supplementary Fig. 1d and e was performed on adult submandibular glands from male mice. We apologize for the confusion and have clarified the text on page 28 line 812 to read “Mass spec analysis of HS disaccharides and tetrasaccharides in adult **male** SMGs. Analysis of HS disaccharide **c**, and tetrasaccharide **d** composition between WT and *Hs3st3* DKO SMGs. All data shown is averaged. N=3 consisting of 3-4 SMGs pooled together for each group.”

We have also included new HS analysis to compare male and female WT and DKO SMGs (n=3 each consisting of 3-4 pooled glands). This new data is included as a part of the supplementary figure 1f and g showing the comparison of data from male and female SMGs from the same batches of mass spectrometry analysis.

The text on page 6, line 150 reads “Direct comparison of male and female SMGs within the same analytical batches showed overall more differences in the HS composition of the disaccharides than the tetrasaccharides between WT males and WT females. Specifically, UA-GlcNAc, UA2S-GlcNAc, and UA2S-GlcNAc6S disaccharides was reduced whereas UA-GlcNAc6S and UA-GlcNS6S was increased in WT female SMGs compared to WT male SMGs (Supplementary Fig. 2f). Only tetrasaccharide UA-2S-GlcNS-IdoA2S-GlcNS3S was different in the SMGs of the two sexes (Supplementary Fig. 2g). The DKO SMGs were more similar in their HS with only UA-2S-GlcNAc6S and UA-2S-GlcNS being different. Further work is needed to fully understand the biological significance of these differences, however, this analysis showed that both males and females have altered HS composition in the SMGs of DKO mice.”

The figure legend for supplementary Fig. 1f and g reads “Analysis of HS disaccharides (f) and tetrasaccharides (g) of adult female and male WT and DKO SMGs. For the analysis, 3-4 glands were pooled together for each sample, 3 samples per group and graphs are showing average for each group (n=3). Unpaired t-test performed. WT female vs WT male: UA-GlcNAc; ** $p=0.0089$, UA2S-GlcNAc; * $p=0.0156$, UA-GlcNAc6S; * $p=0.0101$, UA2S-GlcNAc6S; * $p=0.0261$, UA-GlcNS6S; * $p=0.0216$ and UA2S-GlcNS-IdoA2S-GlcNS3S; * $p=0.0102$. DKO female vs DKO male: Δ UA2S-GlcNAc6S; *** $p=0.0002$ and Δ UA2S-GlcNS; * $p=0.0378$. nd = not detected.”

We apologize for not being clear about the WT controls used for normalization of the qPCR data in Figure 1e. WT controls were sex matched for all adult qPCR. The text for the figure legend on page 28, line 817 now reads “e Gene expression changes in HS biosynthetic enzymes in both male and female DKO SMGs normalized to respective male or female WT SMGs and *Rps29*.”

We also agree with the comment about the sex differences in male and female glands could affect some findings. We have now included the LACE data for the female SMGs in supplementary figure 3. This shows that the binding of the FGF10/FGFR2b-Fc complex is increased in both male and female DKO submandibular glands.

The legend for supplementary Fig. 3 now reads “a Binding of FGF10/FGFR2b-Fc complex overlapping with COLIV, a basement membrane protein, is increased in DKO female SMGs. Representative images of single confocal sections from WT and DKO SMGs showing FGFR2b-Fc (green), COLIV (magenta), and E-cadherin (ECAD, cyan). Scale bar: 10 μ m. b Quantification of A and B protein fluorescence intensity normalized to total nuclei and expressed as a fold change compared to WT. SMGs from five female mice for each genotype were imaged and used for quantification. Error bars: SM. Unpaired t-test compared to WT, LACE ** $p=0.0229$, COLIV * $p=0.0450$.”

The text on page 9 line 225 reads “Surprisingly, the DKO had increased binding of the FGF10/FGFR2b complex surrounding acinar structures and ducts, overlapping with increased type-IV collagen (ColIV) staining in the BM compared to WT (Fig. 3a and supplementary Fig. 3a-b).”

In addition, we have now included data for saliva secretion for male and female separately which shows that the male mice have a significant decrease in saliva flow (~33% reduction), while saliva flow in female mice is not significantly reduced (~17% reduction). The results showing female salivary flow is now included as part of supplementary Fig. 4c.

The text on page 11, line 288 reads “We then measured stimulated whole saliva flow after pilocarpine treatment of both male and female mice. Saliva flow was expressed as a ratio of saliva volume to body weight, the ratio was normalized to WT littermates and there was an overall decrease in saliva flow in male DKO mice (~33% reduction) (Fig. 4g), whereas in female DKO mice, although saliva flow is reduced it is not significant (~17% reduction) (Supplementary Fig. 4c).”

The text for the supplementary figure 4c legend reads “c Salivary flow rates in adult female mice collected after pilocarpine stimulation. Saliva flow normalized to the WT and shown as %. Mean \pm SM. Unpaired t-test, ns.”

2. The Western blot shown in Fig. 3:

There is a lot of background staining with the 3G10 antibody, quite similar in the four lanes. In contrast, according to the b-actin staining, there is much less sample in the heparinase-treated samples. Most importantly, the sizes of the “new” bands obtained after heparinase digestion of the DKO sample do not match any known size of HSPG core proteins. Maybe the band eluting between the 97 and 191 markers could be a syndecan with remaining chondroitin sulfate chains. A combined heparinase/chondroitinase digestion could solve this problem.

Reply: We thank the reviewer for this point and suggestions. The samples without the enzymes were previously not incubated at 37 °C and the differences in β -actin bands most likely are due to proteolytic degradation of the enzyme treated samples or some protein precipitation when lysates are added to the enzyme assay buffer. We apologize for these inconsistencies in the protocol and have now treated all samples with or without heparitinase and chondroitinase ABC under similar conditions. The data included in Figure 3d now show the blots with samples all incubated under similar conditions.

Regarding the band size of the “new bands”, we increased the chondroitinase ABC enzyme concentration, but the sizes of the bands detected after enzyme digestions are still detected at the same molecular weights as previously seen on our blots. We were not able to resolve this problem as it is likely that factors such as the type of SDS-PAGE gels used, the method and type of lysis buffer used to prepare the samples, the enzyme assay buffer, the length of digestion and the type of tissue affect the resolution of these HSPG cores, as compared with other reports using different tissues and methods. The reviewer makes an interesting point about the syndecans that could warrant further investigation, but this is beyond the scope of the project and will require probing the blots with specific antibodies to identify the HSPG cores.

The strong background bands detected in all the lanes indicated by asterisks may represent the light and heavy chains of immunoglobulins present in the salivary gland lysates.

The new data with all samples incubated under similar conditions is now included in figure 3e. The text on page 9 line 243 reads “These data may suggest that HSPGs in the BM are increased in the DKO SMG (Fig. 3c), which is supported by 3G10 Western blot analysis of tissue lysates treated with heparinase III and chondroitinase ABC enzymes, revealing increased high molecular weight core HSPGs (Fig. 3d).”

The figure legend for Figure 3d on page 30, line 863 reads “d Western blot of 3G10 antibody, which detects the HS stub, shows increase in high molecular weight HSPG cores (indicated by red arrows) in DKO male SMG lysates treated with or without heparinase III and chondroitinase ABC. β -actin is also shown. Bands identified with an asterisk may represent the light and heavy chains of immunoglobulins.”

REVIEWERS' COMMENTS

Reviewer #1 (Remarks to the Author):

The authors have addressed my comments, but please revise the sentence on p.6 referred to in the rebuttal letter:

"Specifically, UA-GlcNAc, UA2S- GlcNAc, and UA2S-GlcNAc6S disaccharides was reduced whereas UA-GlcNAc6S and UA- GlcNS6S was increased in WT female SMGs compared to WT male SMGs (Supplementary Fig. 2f)."

Isn't it the other way around? To me it looks like the UA-GlcNAc, UA2S- GlcNAc, and UA2S-GlcNAc6S disaccharides were increased whereas UA-GlcNAc6S and UA- GlcNS6S were reduced in WT female SMGs compared to WT male SMGs. (The fig. is supplementary Fig. 1f)

Ref.: Nature Communications manuscript NCOMMS-24-02038B

Title: Specific 3-*O*-sulfated heparan sulfate domains regulate salivary gland basement membrane metabolism and epithelial differentiation

Response to Reviewers:

We would like to thank reviewer#1 for the helpful suggestion to revise the incorrect statement. We have addressed the concern below, highlighting the changes that have been made to the text in the manuscript.

Reviewer #1 (Remarks to the Author):

The authors have addressed my comments, but please revise the sentence on p.6 referred to in the rebuttal letter:

"Specifically, UA-GlcNAc, UA2S- GlcNAc, and UA2S-GlcNAc6S disaccharides was reduced whereas UA-GlcNAc6S and UA- GlcNS6S was increased in WT female SMGs compared to WT male SMGs (Supplementary Fig. 2f)."

Isn't it the other way around? To me it looks like the UA-GlcNAc, UA2S- GlcNAc, and UA2S-GlcNAc6S disaccharides were increased whereas UA-GlcNAc6S and UA- GlcNS6S were reduced in WT female SMGs compared to WT male SMGs. (The fig. is supplementary Fig. 1f)

Reply: The text has been amended on page 6, line 148 to read "Specifically, UA-GlcNAc, UA2S-GlcNAc, and UA2S-GlcNAc6S disaccharides were increased whereas UA-GlcNAc6S and UA-GlcNS6S were reduced in WT female SMGs compared to WT male SMGs (Supplementary Fig. 1f)."